# FreqKV: Key-Value Compression in Frequency Domain for Context Window Extension

**Jushi Kai[1], Yixuan Wang[1], Boyi Zeng[1], Haoli Bai[2], Bo Jiang[3], Ziwei He[4]\*, Zhouhan Lin[1]\***

[1]LUMIA Lab, Shanghai Jiao Tong University, [2]Huawei Noah's Ark Lab
[3]Shanghai Jiao Tong University, [4]Shanghai Innovation Institute
json.kai@sjtu.edu.cn, ziwei.he@sii.edu.cn, lin.zhouhan@gmail.com

## ABSTRACT

Existing key-value (KV) cache compression methods for large language models (LLMs) often rely on token eviction, which risks losing critical local information in both long prefilling and decoding scenarios. When extrapolating beyond the pretrained context length, their performance degrades sharply on long-context benchmarks. Motivated by the observation in the frequency domain that the context information is concentrated in the low-frequency components, we propose FreqKV, a parameter-free and architecture-agnostic approach. It iteratively compresses the increasing KV cache in the frequency domain, allowing models to process lengthy contexts efficiently. With minimal training at 8K length, FreqKV extends the context window of LLaMA-2-7B up to 256K tokens while maintaining stable perplexity. Extensive experiments across prefilling and decoding demonstrate that FreqKV enables robust context window extension and consistently outperforms existing KV cache compression methods on LLaMA-2 and LLaMA-3, highlighting its effectiveness for both understanding and generation in long contexts. Our code is available at `https://github.com/LUMIA-Group/FreqKV`.

## 1 INTRODUCTION

Large language models (LLMs) have demonstrated remarkable performance in natural language understanding and generation. However, their inference capabilities are fundamentally constrained by a pre-defined context window size established during pre-training. Consequently, LLMs struggle to maintain their performance when processing sequences that exceed the preset context size, significantly limiting their applicability in long-context scenarios.

To alleviate this limitation, several position encoding methods, such as ALiBi (Press et al., 2022), PI (Chen et al., 2023), and LongRoPE (Ding et al., 2024) have been proposed to extend the context window. Relying on full self-attention, the quadratic computational cost renders them prohibitively expensive at scale. LongLoRA (Chen et al., 2024) trains LLMs using shifted sparse attention. Despite training efficiency, their sparse attention fails to be applied during inference, and they still require the original attention on the full sequence.

A popular strategy to improve efficiency is to compress the key-value (KV) cache during inference. Methods such as SnapKV (Li et al., 2024), PyramidKV (Cai. et al., 2024b), and FastKV (Jo et al., 2025) evict tokens deemed less important according to attention scores. However, these compression strategies cannot be extrapolated beyond the context window. The information associated with the evicted tokens will be lost, leading to performance degradation when decoding future tokens.

Another line of work (Zhang et al., 2024; Wan et al., 2025) attempts to merge tokens rather than dropping them, preserving more information. Nevertheless, their performance remains suboptimal without fine-tuning. Concurrently, LoCoCo (Cai et al., 2024) and Activation Beacon (Zhang et al., 2025) introduce additional modules to compress KV states and incorporate the fine-tuned compressing pattern into the decoding procedure of LLMs. However, the increasing memory requirement for additional parameters is inevitable.

---

\*Corresponding authors.

In this work, we observe that the KV states of LLMs exhibit strong energy concentration in the low-frequency components of the frequency domain, suggesting that high-frequency parts are largely redundant. Building on this observation, we introduce FreqKV, an efficient context window extension method that iteratively compresses key-value states in the frequency domain. Unlike eviction methods, FreqKV preserves the information of all tokens by transforming KV states into the frequency domain and discarding high-frequency components that carry limited information. By compressing iteratively as the cache grows, FreqKV retains recent tokens with minimal loss while progressively compressing earlier ones. This parameter-free method requires no architectural modifications and can be applied to both the prefilling stage (long-context understanding) and the decoding stage (long-context language modeling and generation).

Extensive experiments show that FreqKV enables robust context window extension. With minimal fine-tuning at 8K length, FreqKV maintains low perplexity as the evaluation context length scales to 256K. It achieves comparable performance to methods that employ full cache or additional compressors in long context language modeling. Experimental results across LongBench, RULER, and Needle-in-a-Haystack indicate that FreqKV surpasses recently studied KV compression methods in the prefilling stage. Furthermore, FreqKV also delivers substantial improvements in long-generation capability on LongGenBench, validating its effectiveness in decoding scenarios. These results establish frequency-domain compression as a viable approach for extending the context window.

## 2 RELATED WORK

**KV Compression for LLMs.** To alleviate memory and computation costs, a growing body of research explores compressing the KV cache when processing long contexts. One widely studied approach is selective token eviction, which can be traced back to attention sinks (Xiao et al., 2024a;b), suggesting that preserving a small set of initial tokens helps stabilize model performance. Furthermore, MInference (Jiang et al., 2024) propose to accelerate prefilling with unique patterns in long-context attention matrices. More recent methods, SnapKV (Li et al., 2024), InfiniPot (Kim et al., 2024), PyramidKV (Cai. et al., 2024b), GemFilter (Shi et al., 2024), and FastKV (Jo et al., 2025), use attention scores to measure the importance of tokens and discard less important ones. However, LLMs suffer from the permanent loss of the information associated with evicted tokens during decoding. To address this limitation, some researchers introduce cache merging techniques to approximate the original full attention of the existing contexts, such as CaM (Zhang et al., 2024), KVMerger (Wang et al., 2024), and D2O (Wan et al., 2025). Nevertheless, these inference methods often sacrifice performance for efficiency and cannot be extrapolated beyond the context window.

**Efficient Context Extension for LLMs.** Recent advancements in context extension for LLMs have focused on efficiently scaling models to handle longer input sequences without significantly increasing computational costs. LongLoRA (Chen et al., 2024) employs shifted sparse attention during the parameter-efficient fine-tuning. However, this sparse attention mechanism is not applicable during inference, necessitating a return to the original full attention post-training. Other techniques, such as LoCoCo (Cai et al., 2024), integrate convolutional operations into LLMs for compressing long contexts. They fine-tune the compression modules together with LLMs. Landmark attention (Mohtashami & Jaggi, 2023) uses landmark tokens to retrieve previous input blocks. Similarly, Activation Beacon (Zhang et al., 2025) introduces a special token to represent the previous context for compression. However, they need a copy of multi-head attention, which amounts to approximately 2B for 7B models. In contrast, our proposed method achieves context extension without introducing any additional parameters.

**Learning in the Frequency Domain.** Learning in the frequency domain is a well-established technique to compress images and accelerate CNNs (Gueguen et al., 2018). It has been observed that CNNs are more sensitive to low-frequency channels than high-frequency channels (Xu et al., 2020). Moreover, Frequency Principle (Xu et al., 2019; Luo et al., 2019) theoretically shows that low frequencies converge faster than high frequencies for deep neural networks. These works have inspired efforts to process natural language. FNet (Lee-Thorp et al., 2022) improves the efficiency of Transformer encoder architectures by replacing the self-attention layers with the Fourier transform to serve the purpose of mixing tokens. Additionally, Fourier Transformer (He et al., 2023) eliminates redundancies in the context through frequency domain processing within encoder architectures.

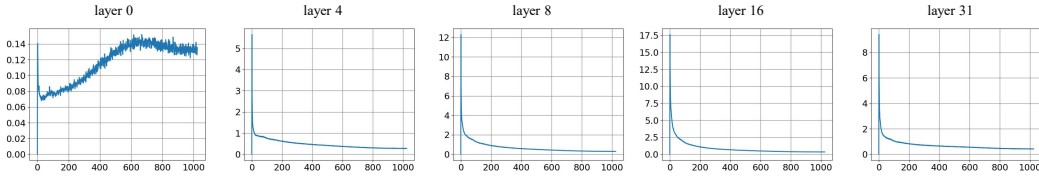

(a) The average power spectrums of key states.

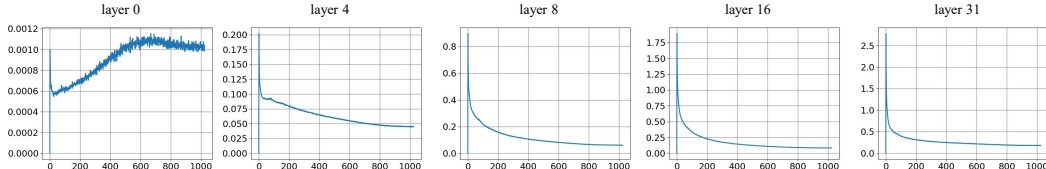

(b) The average power spectrums of value states.

Figure 1: The average power spectrums of key states and value states in different layers of LLaMA-2-7b. 1000 documents are sampled from CNN/Daily Mail (Hermann et al., 2015). We transform key states and value states to the frequency domain, and average power spectrums over these samples and hidden dimensions.

However, because of the autoregressive nature, it remains unclear how to leverage frequency components for decoder-only Transformer, which is the main architecture of generative LLMs. To the best of our knowledge, FreqKV is the first work that explores compressing key-value states in the frequency domain for decoder-only LLMs.

## 3    FREQKV

In this section, we present power spectrum analysis of KV states for LLMs. Building on this observation, FreqKV compresses KV states in the frequency domain and extends the context window via iterative compression.

### 3.1    PRELIMINARIES

**Energy Concentration in the Frequency Domain.**    We use DCT (Discrete Cosine Transform) to transform key states and value states from the time domain, which is the sequence dimension, to the frequency domain. The formulations of DCT and the inverse DCT (IDCT) can be referred to in Appendix A.

The average power spectrums in different decoder layers of LLaMA-2-7B are calculated and presented in Figure 1. The figure reveals that the energy of key states and value states is increasingly concentrated in the low-frequency components. The distributions of their power spectrums are similar since they are projected from the same hidden states. While the initial embeddings of natural languages in the first layer exhibit no strong low-frequency bias, subsequent layers gradually shift energy toward low frequencies along the decoding procedure. This observation suggests that the low-frequency components capture most of the essential information, whereas high-frequency parts carry limited signal and can be discarded with minimal loss. Further analysis of frequency components is provided in Appendix B.

**Self-Attention with KV Cache.**    For the incoming token $x_N$, the prefilled $N$ tokens $\boldsymbol{X}_{0:N-1} = [x_0, \ldots, x_{N-1}]$ are utilized as the cache during decoding. Denote the cached KV states for the previous $N$ tokens $\boldsymbol{X}_{0:N-1}$ as $\boldsymbol{K}_{0:N-1}$ and $\boldsymbol{V}_{0:N-1}$. For simplicity, indices corresponding to layers and heads have been omitted.

When calculating attention scores, the incoming token $x_N$ attends to all cached KV states as well as to itself:

$$\mathcal{A}(N) = \text{Softmax}\left(\frac{\boldsymbol{q}_N[\boldsymbol{K}_{0:N-1} \oplus \boldsymbol{k}_N]^T}{\sqrt{d}}\right) \cdot [\boldsymbol{V}_{0:N-1} \oplus \boldsymbol{v}_N], \tag{1}$$

where $d$ is the hidden dimension. $\boldsymbol{q}_N, \boldsymbol{k}_N$ and $\boldsymbol{v}_N$ are the query, key and value states of $x_N$ respectively. $\oplus$ means the concatenation of the KV cache and KV states of $x_N$.

## 3.2 EFFECT OF FREQUENCY COMPONENTS

To empirically investigate what information different frequency bands encode, we have conducted a targeted perturbation experiment. We sample 100 CNN/DailyMail documents and introduce controlled perturbations by randomly replacing 1% of words with their synonyms. We then transform the KV states of LLaMA-2 into the frequency domain and compare the cosine similarity between perturbed and original inputs for both low-frequency (lowest 50%) and high-frequency (highest 50%) components. The results across different layers are presented in Figure 2:

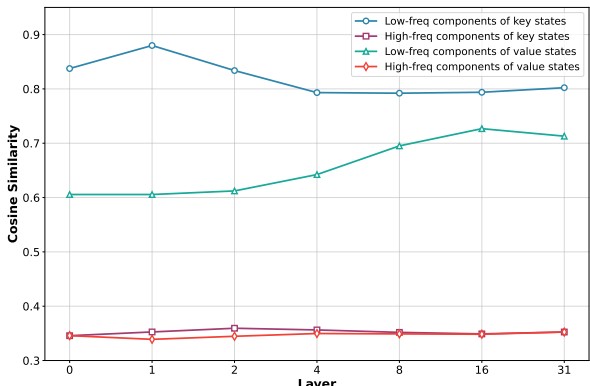

Figure 2: Cosine Similarity between perturbed and original inputs for different frequency components of KV states.

It shows that the output states are robust to the perturbation with low-frequency components retained while high-frequency ones are more sensitive, supporting the interpretation that low frequencies encode global information, while high frequencies capture local details.

We further evaluate this distinction on the summarization track of the downstream benchmark Long-Bench, comparing the performance of llama-2-chat-7b with low and high frequencies retained in Table 1. Summarization requires capturing global semantic structure and long-range coherence. Accordingly, retaining low-frequency components yields substantially better performance.

Table 1: Performance on summarization tasks with high/low-frequency components retained.

| Components Retained | GovReport | QMSum | MultiNews |
|---------------------|-----------|-------|-----------|
| high-frequency | 14.21 | 16.54 | 15.55 |
| low-frequency | **25.51** | **21.81** | **26.9** |

Together, these results strengthen our motivation: low-frequency components primarily encode global semantic contexts and long-range dependencies, whereas high-frequency components encode local details, which explains why preserving low-frequency information is both more robust and more beneficial for long-context understanding.

## 3.3 KV COMPRESSION IN THE FREQUENCY DOMAIN

To reduce redundancy in the key-value (KV) cache, we compress KV states in the frequency domain as shown in Figure 3a. Specifically, we conduct DCT along the sequence dimension to transfer the KV cache to the frequency domain:

$$\boldsymbol{Z}_{0:N-1}^K = \text{DCT}(\boldsymbol{K}_{0:N-1}), \quad \boldsymbol{Z}_{0:N-1}^V = \text{DCT}(\boldsymbol{V}_{0:N-1}), \tag{2}$$

where $\boldsymbol{Z}_{0:N-1}^K$ and $\boldsymbol{Z}_{0:N-1}^V$ are the spectral representations of the KV cache.

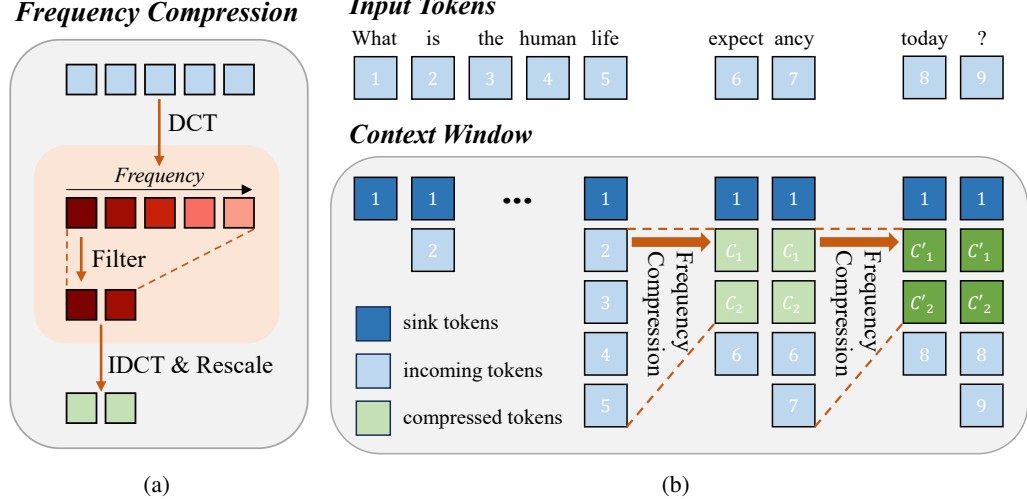

Figure 3: The overview of our FreqKV. (a) The illustration of the frequency-domain compression. (b) The KV cache will be compressed in an iterative manner to extend the context window. Sink tokens remain uncompressed throughout the process. The tokens after sink tokens will be compressed in the frequency domain and subsequent tokens will continue to get into the cache. When the cache is filled again, the compressed tokens and incoming tokens will be compressed together.

As illustrated in Figure 1, since the low-frequency components exhibit higher magnitudes and carry more information, we will retain them and remove the high-frequency parts for compression. Given a retaining ratio $\gamma$, the cache size is reduced to $L = \gamma \cdot N$ by filtering out $N - L$ high-frequency components.

Then we conduct IDCT along the frequency dimension to convert the compressed components back to the time dimension. It should be noted that the signals are normalized by the square root of the component number when applying DCT and IDCT (Appendix A). Therefore, the compressed signals should be rescaled with the factor of $\sqrt{\frac{L}{N}}$ to restore the original amplitude:

$$\widetilde{\boldsymbol{K}}_{0:L-1}^{0:N-1} = \sqrt{\frac{L}{N}}\text{IDCT}(\boldsymbol{Z}_{0:N-1}^{K}[0:L-1]), \quad \widetilde{\boldsymbol{V}}_{0:L-1}^{0:N-1} = \sqrt{\frac{L}{N}}\text{IDCT}(\boldsymbol{Z}_{0:N-1}^{V}[0:L-1]), \quad (3)$$

$\widetilde{\boldsymbol{K}}_{0:L-1}^{0:N-1}$ and $\widetilde{\boldsymbol{V}}_{0:L-1}^{0:N-1}$ are the compressed KV cache of size $L$ in the time domain. The superscript "$0:N-1$" indicates the original token range, while the subscript "$0:L-1$" means the retaining size. The incoming token $x_N$ will attend to the compressed KV cache as follows:

$$\widetilde{\mathcal{A}}(N, L) = \text{Softmax}\left(\frac{\boldsymbol{q}_N[\widetilde{\boldsymbol{K}}_{0:L-1}^{0:N-1} \oplus \boldsymbol{k}_N]^T}{\sqrt{d}}\right) \cdot [\widetilde{\boldsymbol{V}}_{0:L-1}^{0:N-1} \oplus \boldsymbol{v}_N]. \quad (4)$$

### 3.4 CONTEXT EXTENSION VIA ITERATIVE COMPRESSION

Extending the context window of LLMs is fundamentally constrained by memory and computation cost. To address this, FreqKV employs an iterative compression strategy in the frequency domain that constrains the effective cache size while enabling processing of arbitrarily long sequences. The overall pipeline is illustrated in Figure 3b.

For tokens within the context window, the standard attention will be conducted. Once the cache reaches the maximum size, we will compress it in the frequency domain. Specifically, cached KV states are converted to the frequency domain via DCT, where low-frequency components are retained before applying IDCT to transform them back to the time domain. The incoming tokens are appended to the compressed cache, and the process repeats when the cache is filled again. As a result,

earlier tokens experience more iterations of compression as the context window expands, whereas less compression will be performed on the more recent tokens—a property well aligned with the autoregressive nature of LLMs. The compression is triggered only once every $N - L$ tokens, with the complexity of $O(NlogN)$ for each compression. Therefore, the computational overhead of the compression could be negligible.

Recent work has found the phenomenon of attention sinks that LLMs tend to assign high attention scores to initial tokens (Xiao et al., 2024b; Han et al., 2024). Therefore, we maintain these initial tokens uncompressed in the cache and only compress tokens that come after them.

During training and prefilling, FreqKV will process the sentence chunk-wise, interleaving attention computation with the compression operation. After each compression, $N - L - S$ incoming tokens form a chunk that fills the cache, while $S$ sink tokens remain uncompressed, and $L$ compressed "tokens" represent earlier history. Consequently, the newly incoming token $x_M$ will attend to $S$ sink tokens, $L$ compressed tokens, $M - N$ previous incoming tokens, and itself:

$$\widetilde{\mathcal{A}}(S, N, L, M) = \text{Softmax}\left(\frac{\boldsymbol{q}_M[\boldsymbol{K}_{0:S-1} \oplus \widetilde{\boldsymbol{K}}_{0:L-1}^{S:N-1} \oplus \boldsymbol{K}_{N:M}]^T}{\sqrt{d}}\right) \cdot [\boldsymbol{V}_{0:S-1} \oplus \widetilde{\boldsymbol{V}}_{0:L-1}^{S:N-1} \oplus \boldsymbol{V}_{N:M}], \quad (5)$$

where $\widetilde{\boldsymbol{K}}_{0:L-1}^{S:N-1}$ and $\widetilde{\boldsymbol{V}}_{0:L-1}^{S:N-1}$ denote the compressed KV cache of size $L$ for tokens $\boldsymbol{X}_{S:N-1} = [x_S, \ldots, x_{N-1}]$.

It should be noted that key states in LLMs are often equipped with position embeddings like RoPE (Su et al., 2024). In FreqKV, key states are compressed and cached before applying RoPE. The cached key states are then encoded with RoPE when conducting self-attention. They are allocated with position indices within the cache rather than the original sequence, which results in the context window extension without requiring position extrapolation or interpolation. Further discussion of RoPE integration is provided in Appendix C.

## 4 EXPERIMENTS

### 4.1 EXPERIMENTAL SETTINGS

We employ FreqKV on LLaMA-2-7B/13B (Touvron et al., 2023) and LLaMA-3-8B (AI@Meta, 2024). Unless otherwise specified, the retaining ratio $\gamma$ is set to 0.5, and $S = 4$ sink tokens are kept uncompressed. For LLaMA-2, the preset context window size is 4096, which also defines the maximum KV cache size $N$. Accordingly, during each compression step, the $N - S = 4092$ states since the 5-th state in the cache will be compressed into $L = \gamma \cdot (N - S) = 2046$ states. As for LLaMA-3, the native context window size is 8192. It is equipped with GQA (Grouped-Query Attention), which means it has a lower proportion of parameters for attention modules than LLaMA-2 (Multi-Head Attention, MHA). Minimal training is introduced to adapt models to this frequency-domain compression method, following LongLoRA (Chen et al., 2024). Details of training settings are summarized in Appendix D.

We assess FreqKV on long context language modeling and generation tasks for decoding performance, and long context understanding tasks for prefilling performance. For long context language modeling, we fine-tune LLaMA-2-base on the pre-training dataset RedPajama (Computer, 2023), extending the context window from 4K to 32K. Perplexity (PPL) evaluation is conducted on PG-19 (Rae et al., 2019). For long context generation and understanding, the instruction following dataset LongAlpaca (Chen et al., 2024) is used for the supervised fine-tuning (SFT) of LLaMA-2-chat and LLaMA-3-instruct. The context window is extended from 4K to 8K, and from 8K to 16K respectively. Models are evaluated on LongGenBench (Liu et al., 2024), LongBench (Bai et al., 2024), RULER (Hsieh et al., 2024), and Needle-in-a-Haystack (gkamradt, 2023).

### 4.2 LONG CONTEXT LANGUAGE MODELING

We use FreqKV to train LLaMA-2-7B and LLaMA-2-13B on RedPajama. Perplexity is measured on the test set of the book corpus dataset PG-19 with the evaluation sliding window of 256. We compare our method with other baselines, including full fine-tuning (Full FT), LongLoRA, and

Table 2: Perplexity evaluation on the test set of PG-19. The superscript "*" means that we reproduce LoCoCo following their official code for evaluation. The results of full fine-tuning and LongLoRA are reported from LongLoRA (Chen et al., 2024). Full FT and LoCoCo encounter the OOM (Out-of-Memory) issue with the training length of 16K.

| Size | Training Length | Method | Inference Cache | Evaluation Context Length | | | | |
|------|-----------------|--------|-----------------|------|------|------|-------|-------|
| | | | | 2048 | 4096 | 8192 | 16384 | 32768 |
| 7B | 8192 | Full FT | Full | 7.55 | 7.21 | 6.98 | - | - |
| | | LoCoCo* | Compressed | 8.15 | 8.08 | 7.27 | - | - |
| | | LongLoRA | Full | 7.70 | 7.35 | 7.14 | - | - |
| | | **FreqKV** | Compressed | **7.45** | **7.12** | **7.04** | **7.02** | **7.02** |
| | 16384 | Full FT | Full | | OOM during Training | | | |
| | | LoCoCo | Compressed | | OOM during Training | | | |
| | | LongLoRA | Full | 7.65 | 7.28 | **7.02** | **6.86** | - |
| | | **FreqKV** | Compressed | **7.46** | **7.13** | 7.03 | 6.99 | **6.98** |
| | 32768 | LongLoRA | Full | 8.29 | 7.83 | 7.54 | 7.35 | 7.22 |
| | | **FreqKV** | Compressed | **7.47** | **7.14** | **7.04** | **7.00** | **6.98** |
| 13B | 8192 | Full FT | Full | 6.95 | 6.60 | 6.43 | - | - |
| | | LongLoRA | Full | 7.03 | 6.73 | 6.58 | - | - |
| | | **FreqKV** | Compressed | **6.82** | **6.52** | **6.44** | **6.42** | **6.41** |

LoCoCo (Cai et al., 2024). While Full FT and LongLoRA leverage full KV cache during inference, LoCoCo and FreqKV use compressed cache.

PPL scores on different evaluation context lengths are reported in Table 2. More Evaluation results on Proof-pile (Azerbayev et al., 2022) are presented in Appendix E. While both LongLoRA and LoCoCo sacrifice performance within the original context length (2K and 4K), FreqKV matches or even surpasses Full FT. With the training length of 8K, FreqKV performs comparably to LongLoRA trained at 32K and consistently outperforms LoCoCo in terms of extended context length as well as the shorter lengths. Additional ablation studies on retained frequency components, retaining ratio, and sink size are reported in Appendix F.

## 4.3 LONG CONTEXT UNDERSTANDING

To further assess the performance on long-prefilling scenarios, we apply FreqKV to extend the context window of LLaMA-2-chat-7b from 4K to 8K and LLaMA-3-instruct-8b from 8K to 16K. Evaluation is conducted on three long context understanding benchmarks: LongBench, RULER, and Needle-in-a-Haystack.

**LongBench.** Scores on the 5 categories of LongBench are reported in Table 3. FreqKV consistently improves the performance of LLaMA-2-chat and LLaMA-3-instruct across a majority of these tasks. We compare our method against *eviction-based* strategies GemFilter (Shi et al., 2024), SnapKV (Li et al., 2024), PyramidKV (Cai. et al., 2024b), LaCache (Shi et al., 2025), and FastKV (Jo et al., 2025), *merging-based* strategies CaM (Zhang et al., 2024), KVMerger (Wang et al., 2024), and D2O (Wan et al., 2025), and a channel compression method ThinK (Xu et al., 2025) that improves SnapKV. FreqKV achieves SOTA (state-of-the-art) on most of the long context understanding tasks and obtains the highest average score. In particular, it shows significant gains on question answering and summarization tasks such as HotpotQA, 2WikiMQA, and QMSum. Moreover, even with an extremely low retaining ratio ($\gamma = 1\%$), where other methods degrade sharply, FreqKV remains stable and effective, highlighting its robustness under aggressive compression.

**RULER.** Results on RULER are presented in Table 4. FreqKV consistently outperforms baselines across different tasks. Notably, baseline methods collapse entirely at 16K evaluation length, which exceeds the original context window of LLaMA-3-instruct (8K). This failure arises because they

Table 3: Scores of different KV compression methods on LongBench. The superscript "*" means that results of CaM and KVMerger are reported from KVMerger (Wang et al., 2024). We reproduce the other baselines following their official codes for evaluation. The second-best scores are also underlined in the table.

| Retaining Ratio | Method | Single-Document QA | | | Multi-Document QA | | | Summarization | | | Few-shot Learning | | | Code | | Avg. |
|---|---|---|---|---|---|---|---|---|---|---|---|---|---|---|---|---|
| | | NtrvQA | Qasper | MF-en | HotpotQA | 2WikiMQA | Musique | GovReport | QMSum | MultiNews | TREC | TriviaQA | SAMSum | LCC | RB-P | |
| *LLaMA-2-chat-7B* | | | | | | | | | | | | | | | | |
| 100% | Full Cache | 18.7 | 19.2 | 36.8 | 25.4 | 32.8 | 9.4 | 27.3 | 20.8 | 25.8 | 61.5 | 77.8 | 40.7 | 52.4 | 43.8 | 35.17 |
| 50% | CaM* | 17.08 | 20.00 | 33.98 | - | 30.69 | - | 24.46 | - | 24.66 | 63.5 | 82.17 | - | - | - | - |
| | KVMerger* | 18.50 | 20.04 | 36.89 | - | 32.99 | - | 25.31 | - | 26.29 | 64.0 | 83.62 | - | - | - | - |
| | LaCache | 16.27 | 16.55 | 31.55 | 32.62 | 26.22 | 7.72 | 22.84 | 20.34 | 25.16 | 65.0 | 83.28 | 38.46 | 47.97 | 43.41 | 34.10 |
| | D2O | 20.32 | 19.16 | 33.09 | 30.62 | 26.76 | 9.65 | 23.72 | 20.54 | 25.42 | 63.0 | 84.23 | 41.22 | 58.61 | 52.77 | 36.37 |
| | GemFilter | 15.22 | 17.76 | 32.92 | 27.29 | 26.76 | 8.66 | 16.17 | 16.29 | 15.95 | 57.5 | 84.8 | 31.96 | 46.54 | 45.14 | 31.64 |
| | FastKV | 18.25 | 22.46 | 35.83 | 28.84 | 31.34 | 8.49 | 24.4 | 20.96 | 25.8 | 60.0 | 82.67 | 40.64 | 57.43 | 51.52 | 36.33 |
| | SnapKV | 17.91 | 22.9 | 35.3 | 26.03 | 28.21 | 8.77 | 24.99 | 20.94 | 25.92 | 64.0 | 83.36 | 41.0 | 60.83 | 55.14 | 36.81 |
| | ThinK | 15.57 | 21.76 | 37.98 | 30.21 | 31.55 | 8.06 | 26.24 | 20.98 | 26.1 | 65.0 | 80.08 | 40.97 | 60.2 | 54.26 | 37.07 |
| | PyramidKV | 17.36 | 23.58 | 34.89 | 26.28 | 28.39 | 9.17 | 24.95 | 20.9 | 26.04 | 64.0 | 83.36 | 40.92 | 60.68 | 55.17 | 36.84 |
| | **FreqKV** | **20.41** | 21.05 | 31.2 | **34.54** | **34.78** | **14.47** | 25.51 | **21.81** | **26.9** | 56.0 | 84.09 | **41.53** | 58.99 | **58.65** | **37.85** |
| 1% | LaCache | 4.37 | 12.72 | 6.34 | 14.77 | 14.91 | 2.23 | 11.18 | 7.39 | 9.94 | 16.75 | 15.13 | 6.42 | 7.56 | 7.09 | 9.77 |
| | D2O | 15.53 | 12.61 | 17.64 | 29.0 | 21.71 | 8.49 | 16.56 | 16.05 | 16.63 | 52.5 | 81.03 | 20.2 | 31.88 | 33.47 | 26.66 |
| | FastKV | 10.55 | 13.49 | 13.53 | 13.27 | 6.8 | 4.11 | 11.94 | 14.43 | 11.92 | 27.0 | 52.05 | 17.22 | 32.1 | 29.38 | 18.41 |
| | GemFilter | 2.36 | 3.22 | 3.99 | 1.28 | 2.09 | 0.78 | 11.07 | 13.14 | 10.9 | 12.5 | 36.38 | 15.54 | 21.16 | 26.61 | 11.50 |
| | SnapKV | 11.49 | 12.47 | 13.61 | 12.79 | 7.05 | 3.48 | 11.96 | 14.61 | 12.16 | 28.0 | 52.11 | 17.03 | 32.48 | 28.39 | 18.40 |
| | ThinK | 8.38 | 14.5 | 14.71 | 20.13 | 22.21 | 4.94 | 11.97 | 16.44 | 11.63 | 18.0 | 19.59 | 9.46 | 21.87 | 17.09 | 15.07 |
| | PyramidKV | 18.17 | 23.8 | 35.97 | 28.42 | 31.7 | 8.76 | 15.85 | 16.64 | 16.2 | 64.5 | 83.65 | 32.41 | 60.38 | 54.89 | 35.10 |
| | **FreqKV** | **19.44** | 20.2 | 31.68 | **35.89** | 32.0 | **12.52** | 15.96 | **17.39** | 15.91 | 60.5 | 83.98 | 33.07 | 62.47 | 56.57 | 35.54 |
| *LLaMA-3-instruct-8B* | | | | | | | | | | | | | | | | |
| 100% | Full Cache | 22.52 | 31.83 | 41.04 | 44.14 | 36.68 | 24.05 | 28.87 | 23.25 | 26.46 | 75.5 | 90.23 | 42.09 | 56.7 | 51.84 | 42.51 |
| 50% | LaCache | 22.95 | 28.91 | 40.99 | 46.34 | 36.26 | 23.02 | 16.92 | 16.02 | 15.59 | 74.0 | 91.11 | 32.97 | 55.09 | 49.31 | 39.25 |
| | D2O | 26.61 | 28.29 | 38.85 | 44.95 | 36.57 | 22.9 | 27.66 | 23.34 | 25.62 | 74.0 | 90.31 | 41.79 | 57.57 | 54.54 | 42.36 |
| | GemFilter | 23.6 | 29.26 | 43.71 | 42.18 | 36.35 | 24.48 | 17.75 | 16.46 | 15.7 | 73.0 | 90.79 | 33.3 | 53.19 | 38.54 | 38.45 |
| | FastKV | 23.88 | 29.67 | 40.75 | 44.66 | 34.02 | 23.07 | 16.86 | 16.45 | 15.61 | 74.0 | 90.31 | 42.02 | 59.21 | 53.3 | 39.63 |
| | SnapKV | 22.99 | 31.89 | 41.11 | 43.92 | 36.42 | 23.38 | 17.14 | 16.45 | 15.53 | 74.5 | 90.23 | 33.72 | 59.03 | 53.97 | 40.02 |
| | ThinK | 21.98 | 30.97 | 41.9 | 45.1 | 34.18 | 22.51 | 27.73 | 23.38 | 26.4 | 74.5 | 90.16 | 41.5 | 62.05 | 58.37 | 42.91 |
| | PyramidKV | 22.27 | 31.93 | 41.03 | 44.79 | 36.56 | 23.86 | 28.77 | 23.14 | 26.62 | 74.5 | 90.25 | 42.53 | 59.11 | 53.33 | 42.76 |
| | **FreqKV** | 23.14 | **41.6** | 43.62 | **46.48** | **40.54** | 20.2 | **29.99** | **23.67** | **28.58** | 69.0 | 88.95 | 41.47 | 60.33 | 51.11 | **43.48** |
| 1% | LaCache | 14.27 | 5.87 | 20.73 | 40.86 | 33.07 | 18.58 | 15.01 | 7.76 | 10.89 | 40.0 | 77.92 | 16.23 | 40.7 | 40.51 | 27.31 |
| | D2O | 24.95 | 12.77 | 20.19 | 36.61 | 27.2 | 15.43 | 22.57 | 15.18 | 15.12 | 56.0 | 89.34 | 24.75 | 32.79 | 36.18 | 30.65 |
| | GemFilter | 11.8 | 6.48 | 19.3 | 15.23 | 15.35 | 7.8 | 12.28 | 14.25 | 9.84 | 53.42 | 54.19 | 20.51 | 19.37 | 21.79 | 20.12 |
| | FastKV | 19.57 | 12.61 | 31.28 | 42.59 | 29.44 | 20.71 | 13.96 | 15.32 | 12.86 | 61.0 | 87.79 | 29.55 | 56.74 | 54.18 | 34.83 |
| | SnapKV | 18.29 | 14.25 | 31.87 | 38.03 | 26.82 | 20.03 | 14.29 | 14.57 | 13.08 | 60.5 | 88.52 | 29.95 | 55.89 | 53.36 | 34.25 |
| | ThinK | 18.02 | 10.38 | 31.46 | 39.1 | 26.08 | 17.46 | 17.31 | 21.48 | 17.97 | 39.5 | 88.18 | 37.27 | 54.69 | 55.19 | 33.86 |
| | PyramidKV | 22.79 | 31.91 | 41.24 | 44.2 | 36.67 | 22.53 | 17.29 | 16.52 | 15.57 | 74.5 | 90.23 | 33.3 | 58.95 | 53.12 | 39.92 |
| | **FreqKV** | 22.86 | **39.7** | 36.44 | **44.29** | **39.59** | **22.57** | **29.1** | **22.92** | **28.35** | 65.0 | 89.0 | **41.9** | **62.17** | **55.42** | **42.81** |

compress the long prompt after the first decoding step. Out-of-bound position embeddings are used when calculating attention, resulting in a performance breakdown. In contrast, FreqKV remains robust and demonstrates its effectiveness for context extension. FreqKV first conducts self-attention within the context window, and then compresses the KV states for out-of-window tokens to get in. Compressing key states before applying RoPE, FreqKV extends the context window without requiring position extrapolation or interpolation.

**Needle-in-a-Haystack.** Figure 4 illustrates evaluation results on Needle-in-a-Haystack. FreqKV performs well in extending the context window of LLaMA-3-instruct. In contrast, other KV compression methods fail beyond the original window boundary. This breakdown is consistent with our observations in RULER.

## 4.4 Long Context Generation

We use LongGenBench to evaluate the long-context generation performance of FreqKV. This benchmark constructs extended generation tasks by concatenating multiple chain-of-thought questions and answers, followed by several real questions in GSM8K and MMLU for long-context responses. The model is then required to generate answers with a maximum output length of 4096 tokens.

Table 4: Scores of different KV compression methods on RULER. D2O encounters the OOM issue with the evaluation length of 16K.

| Evaluation Length | Method | CWE | FWE | QA | VT | NIAH | Avg. |
|---|---|---|---|---|---|---|---|
| 8K | LLaMA-3 | 96.40 | 82.33 | 62.50 | 95.00 | 97.81 | 86.81 |
| | +D2O | 93.20 | 81.00 | 62.00 | 88.80 | 47.53 | 74.51 |
| | +GemFilter | **96.40** | 82.33 | 62.50 | 95.00 | **97.81** | 86.81 |
| | +FastKV | 86.10 | 77.67 | 63.50 | 93.00 | 87.03 | 81.46 |
| | +SnapKV | 96.00 | 80.33 | 62.50 | 93.60 | 92.28 | 84.94 |
| | +ThinK | 95.7 | 78.67 | 62.00 | 96.8 | 94.91 | 85.62 |
| | +PyramidKV | **96.40** | 82.33 | 62.50 | 95.00 | **97.81** | 86.81 |
| | **+FreqKV** | 91.80 | **84.67** | 64.50 | 100.00 | 97.44 | **87.68** |
| 16K | +D2O | - | - | - | - | - | OOM |
| | +GemFilter | 0.00 | 0.00 | 0.50 | 0.00 | 0.00 | 0.10 |
| | +FastKV | 0.00 | 0.00 | 0.00 | 0.00 | 0.00 | 0.00 |
| | +SnapKV | 0.00 | 0.00 | 0.50 | 0.00 | 0.00 | 0.10 |
| | +ThinK | 0.00 | 0.00 | 0.00 | 0.00 | 0.00 | 0.00 |
| | +PyramidKV | 0.00 | 0.00 | 0.50 | 0.00 | 0.00 | 0.10 |
| | **+FreqKV** | **45.10** | **87.33** | **34.00** | **44.60** | **25.47** | **47.30** |

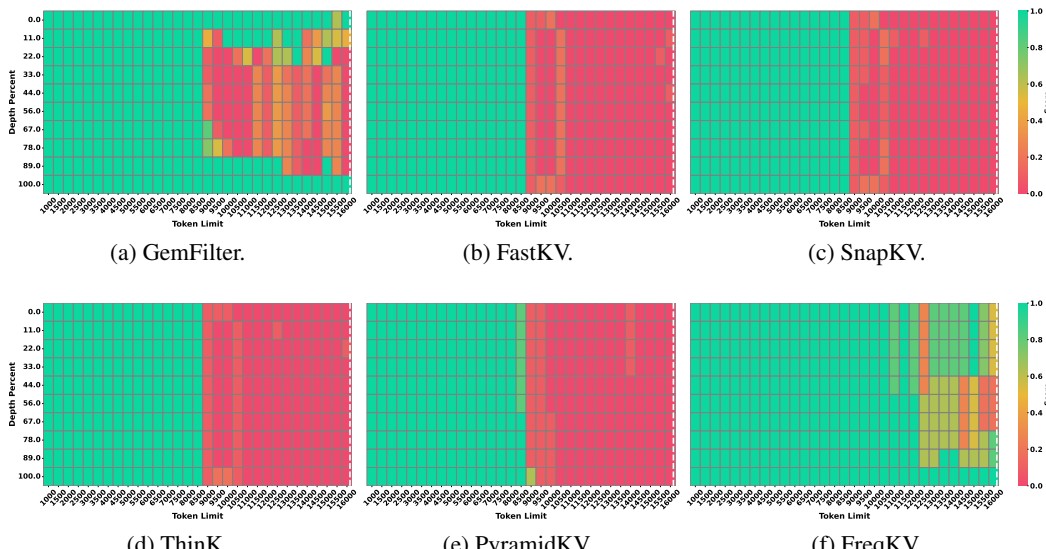

(a) GemFilter.  (b) FastKV.  (c) SnapKV.

(d) ThinK.  (e) PyramidKV.  (f) FreqKV.

Figure 4: The Needle-in-a-Haystack results of different KV compression methods on LLaMA-3-instruct-8B, with the x-axis representing the document length ranging from 1K to 16K tokens, and the y-axis showing the position of the "needle" within the document.

The accuracy scores (%) are provided in Table 5. FreqKV demonstrates substantial improvements in long-generation capability and consistently outperforms existing KV compression methods across both GSM8K and MMLU. FreqKV preserves essential contextual information through iterative frequency-domain compression, enabling robust long-context decoding.

Table 5: Performance on LongGenBench.

| Method | GSM8K | MMLU |
|---|---|---|
| LLaMA-3 | 42.8 | 23.8 |
| +D2O | 8.0 | 24.8 |
| +GemFilter | 48.5 | 25.4 |
| +FastKV | 48.5 | 25.4 |
| +SnapKV | 49.6 | 25.8 |
| +ThinK | 45.5 | 25.7 |
| +PyramidKV | 49.6 | 26.2 |
| **+FreqKV** | **54.2** | **31.2** |

## 5 ANALYSIS

### 5.1 EFFICIENT EXTRAPOLATION

Furthermore, we evaluate the performance of FreqKV at longer lengths on the validation set of PG-19 as

Table 6: Extrapolation performance of FreqKV on PG-19 up to 256K context length. We measure latencies (seconds) and PPL scores at different lengths. "Compression Count" denotes the number of compressions during decoding. "Compression Latency" stands for the latency introduced by all the frequency-domain compressions during decoding.

| Evaluation Length | 8K | 16K | 32K | 64K | 128K | 256K |
|---|---|---|---|---|---|---|
| Compression Count | 3 | 7 | 15 | 31 | 63 | 127 |
| Decoding Latency | 214.96 | 445.69 | 907.49 | 1832.77 | 3692.03 | 7422.64 |
| Compression Latency | 0.62 (0.29%) | 1.09 (0.24%) | 2.18 (0.24%) | 4.00 (0.22%) | 8.31 (0.23%) | 16.39 (0.22%) |
| PPL | 7.76 | 7.74 | 7.73 | 7.73 | 7.73 | 7.73 |

Table 7: The training time and memory requirements of FreqKV and Full FT.

| Method | Training Length | Time (hours) | Memory (GB) |
|---|---|---|---|
| Full FT | 8K | 28.88 | 44.04 |
| FreqKV | 8K | 17.07 | 24.58 |
| | 16K | 38.57 | 34.00 |
| | 32K | 89.35 | 45.37 |

shown in Table 6. With the training length limited to
8K tokens, it effectively extrapolates the context window of LLaMA-2-7B to 256K, which is 64 times larger than the original size. This is enabled by its iterative compression strategy, which applies fewer compressions to recent tokens while progressively compressing earlier ones. Consequently, FreqKV maintains low perplexity as the evaluation context length scales from 8K to 256K, far exceeding the training length.

In addition, latencies (seconds) of decoding and frequency-domain compression at different lengths are also reported in Table 6. It shows that the decoding time of FreqKV increases approximately linearly, with a negligible time spent on compression. The overhead of the compression accounts for less than 0.3% and does not grow with the context length since the number of compressions increases linearly with the context length. Further analysis for the compression overhead is provided in Appendix G. In summary, FreqKV demonstrates strong extrapolation ability to ultra-long contexts while preserving both efficiency and stability.

## 5.2 TRAINING COST

The training time and memory requirements of FreqKV and Full FT on LLaMA-2-7B are reported in Table 7. 8 NVIDIA RTX6000 Ada-48GB GPUs are used. For comparison, we set the *batch size per device* to 1 and *gradient accumulation steps* to 8. Statistics show that FreqKV significantly reduces both training time (by 40.89%) and memory requirements (by 44.19%), demonstrating its computational efficiency.

## 6 CONCLUSION

In this paper, we introduce FreqKV, a frequency-domain KV cache compression framework that exploits the energy concentration of KV states to retain low-frequency components while discarding redundant high-frequency ones for LLMs. By iteratively compressing the KV cache, FreqKV achieves efficient long-context extension while preserving decoding quality. Without introducing additional parameters or architectural modifications, FreqKV exhibits remarkable efficiency and effectiveness on both prefilling and decoding stages. Extensive experiments across language modeling, generation, and understanding benchmarks demonstrate that FreqKV not only consistently surpasses existing KV compression methods but also offers a viable and efficient approach for context window extension.

## ETHICS STATEMENTS

Our work pertains to key-value compression and context extension of large language models. In this work, we use only publicly available data and artifacts. There are no ethical issues in our paper, including its motivation and experiments.

## REPRODUCIBILITY STATEMENTS

We have provided detailed implementations of our method throughout the paper. Our method is comprehensively elaborated in Section 3. Detailed settings of our experiments and analyses are given in Section 4, 5, and Appendix F. Our code is included in the Supplementary Material for reproducibility.

## THE USE OF LARGE LANGUAGE MODELS

We used ChatGPT solely as a language editing tool to polish grammar, improve clarity, and refine the academic style of the manuscript. All research ideas, methods, experiments, and analyses were independently developed and conducted by the authors.

## ACKNOWLEDGEMENT

This work is sponsored by the National Natural Science Foundation of China (NSFC) grant (No. 62576211) and the National Key Research and Development Program of China (No. 2023ZD0121402).

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

## A DISCRETE COSINE TRANSFORM

The Discrete Cosine Transform (DCT) transforms a signal from the spatial domain (time or position) into the frequency domain. Several variants of DCT exist, with DCT-II being the most common. For a real-value discrete signal $\boldsymbol{X}_{0:N-1} = [x_0, \ldots, x_{N-1}]$ of length $N$, it is defined as:

$$z_t = \alpha_t \sum_{n=0}^{N-1} x_n \cdot \cos\left[\frac{\pi t(2n+1)}{2N}\right], \quad \alpha_t = \begin{cases} \sqrt{\frac{1}{N}} & \text{if } t = 0, \\ \sqrt{\frac{2}{N}} & \text{otherwise} \end{cases} \quad (6)$$

where $t = 0, 1, \cdots, N-1$. $\alpha_t$ is the normalization factor. The time-domain signal $\boldsymbol{X}_{0:N-1}$ can be recovered by applying the inverse DCT (IDCT) on the frequency components $\boldsymbol{Z}_{0:N-1}$:

$$x_n = \sum_{t=0}^{N-1} \alpha_t \cdot z_t \cdot \cos\left[\frac{\pi t(2n+1)}{2N}\right]. \quad (7)$$

The frequency components are expressed as a combination of the original signals. The values can be computed using the Fast Fourier Transform (FFT) with a complexity of $O(N\log N)$. The amplitudes of frequency components are utilized in the power spectrum analysis to represent the energy or magnitude of components. The components of higher energy in the frequency domain indicate they are more informative (He et al., 2023).

For time signals (e.g., texts) and space signals (e.g., images), their energy is most concentrated in low-frequency DCT components. This phenomenon occurs due to the mathematical properties of DCT and the inherent smoothness of real-world signals. The DCT expresses a signal as a sum of cosine functions oscillating at different frequencies. In the case of text embeddings, neighboring tokens exhibit strong temporal correlations, meaning adjacent values in the semantic space tend to be similar. Low-frequency cosine components are slowly varying and capture these smooth variations well, while high-frequency components mainly represent abrupt changes or noise.

## B HEAD-WISE ANALYSIS OF POWER SPECTRUMS

We explore the power spectrum distribution of key states and value states in different attention heads of LLaMA-2-7B, as shown in Figure 5. Although the values of power spectrums vary in different heads, their distribution exhibits similar patterns. They have a consistent tendency to aggregate energy in the low-frequency components along the decoding procedure. It could be promising to study specific differences and associations in different heads or other modules.

## C COMPRESSING KEY STATES WITH ROPE

We further investigate to compress key states after applying RoPE. Power spectrums of key states with RoPE are provided in Figure 6. Components of certain frequencies are filtered out by RoPE.

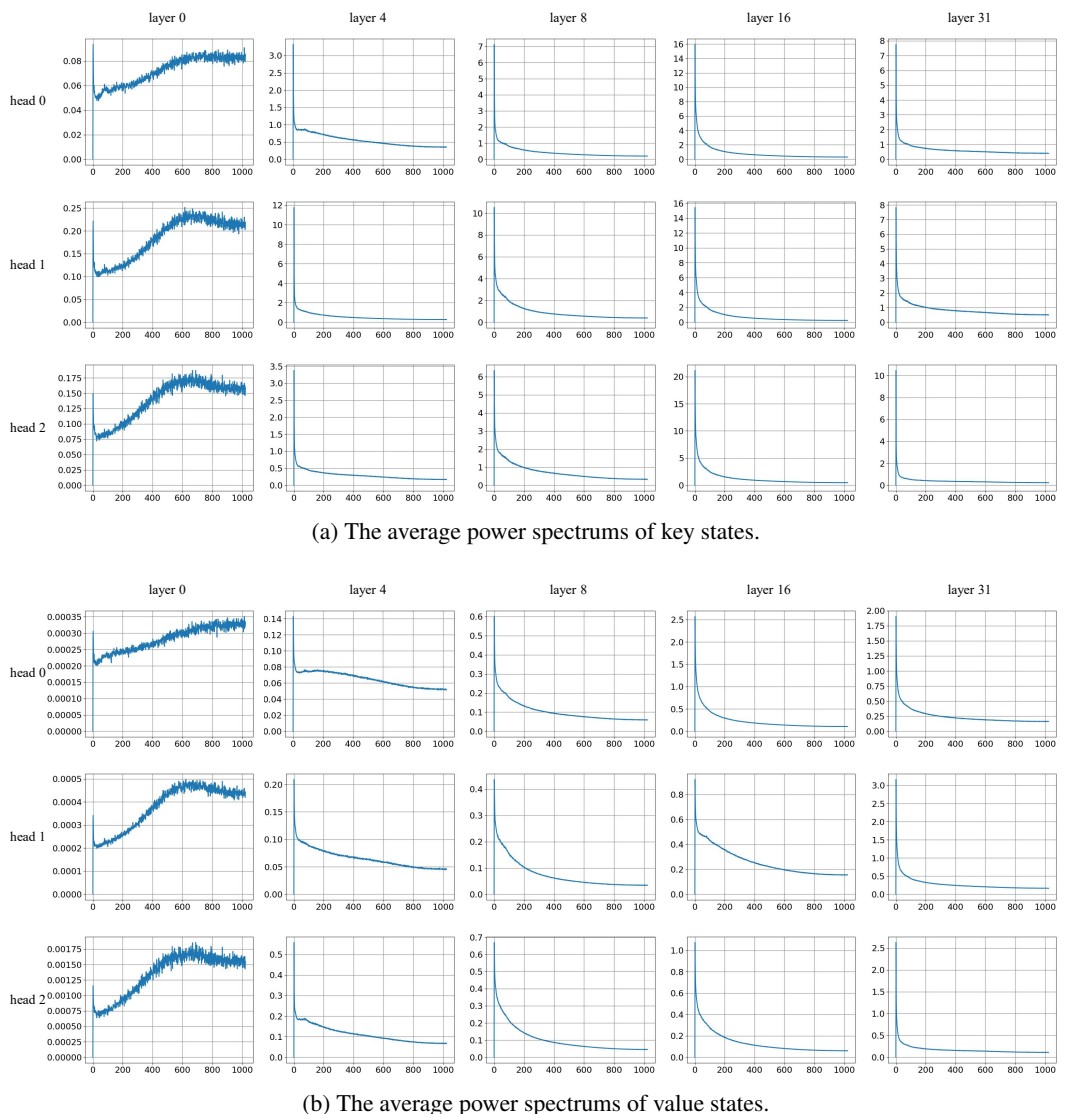

(a) The average power spectrums of key states.

(b) The average power spectrums of value states.

Figure 5: The average power spectrums of key states and value states in different heads of Llama-2-7b.

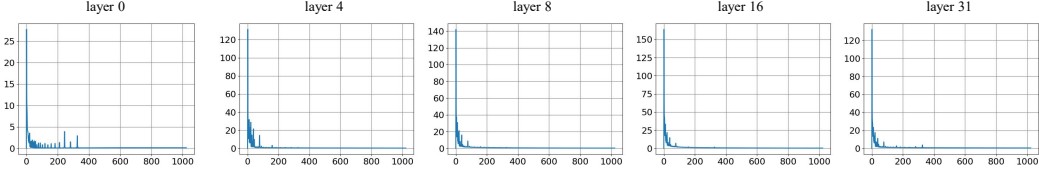

Figure 6: The average power spectrums of key states with RoPE.

We train LLaMA-2-7B on RedPajama with the variant method that key states are compressed after applying RoPE. The training length is 8K. Models are evaluated on the test set of PG-19 and Proof-pile. The results in Table 8 demonstrate that the variant suffers from performance degradation, which is probably due to the usage of position embeddings out of the original window. In contrast, FreqKV compresses and caches key states before applying RoPE. Therefore, they are allocated with position indices within the original window rather than the lengthy sequence, which enables context window extension with no need of using out-of-bound position embeddings.

Table 8: Perplexity evaluation on the test sets of PG-19 and Proof-pile. "Variant" stands for FreqKV with key states compressed after applying RoPE.

| Dataset | Method | Evaluation Context Length | | |
|---|---|---|---|---|
| | | 2048 | 4096 | 8192 |
| *PG-19* | Full FT | 7.55 | 7.21 | 6.98 |
| | FreqKV | 7.45 | 7.12 | 7.04 |
| | Variant | 7.53 | 7.19 | 7.13 |
| *Proof-pile* | Full FT | 3.14 | 2.85 | 2.66 |
| | FreqKV | 3.15 | 2.88 | 2.79 |
| | Variant | 3.16 | 2.88 | 2.80 |

Table 9: Perplexity evaluation on the test set of Proof-pile. The superscript "*" means that we reproduce LoCoCo following their official code for evaluation. The results of full fine-tuning and LongLoRA are reported from LongLoRA (Chen et al., 2024). Full FT and LoCoCo encounter the OOM (Out-of-Memory) issue with the training length of 16K.

| Size | Training Length | Method | Inference Cache | Evaluation Context Length | | | | |
|---|---|---|---|---|---|---|---|---|
| | | | | 2048 | 4096 | 8192 | 16384 | 32768 |
| 7B | 8192 | Full FT | Full | 3.14 | 2.85 | 2.66 | - | - |
| | | LoCoCo | Compressed | 3.40 | 3.20 | 2.88 | - | - |
| | | LongLoRA | Full | 3.20 | 2.91 | **2.72** | - | - |
| | | **FreqKV** | Compressed | **3.15** | **2.88** | 2.79 | **2.76** | **2.75** |
| | 16384 | Full FT | Full | OOM during Training | | | | |
| | | LoCoCo | Compressed | OOM during Training | | | | |
| | | LongLoRA | Full | 3.17 | **2.87** | **2.66** | **2.51** | - |
| | | **FreqKV** | Compressed | **3.15** | 2.88 | 2.78 | 2.74 | **2.73** |
| | 32768 | LongLoRA | Full | 3.35 | 3.01 | **2.78** | **2.61** | **2.50** |
| | | **FreqKV** | Compressed | **3.16** | **2.89** | 2.78 | 2.74 | 2.72 |
| 13B | 8192 | Full FT | Full | 2.96 | 2.69 | 2.53 | - | - |
| | | LongLoRA | Full | 3.04 | 2.77 | **2.60** | - | - |
| | | **FreqKV** | Compressed | **2.99** | **2.74** | 2.65 | **2.63** | **2.63** |

# D  TRAINING SETTINGS

For long context language modeling, we fine-tune LLaMA-2-base on the RedPajama (Computer, 2023) pre-training dataset for 1000 steps. For long context understanding, we SFT LLaMA-2-chat and LLaMA-3-instruct on the instruction following dataset LongAlpaca (Chen et al., 2024) for 5 epochs, which consists of 6.28K long-context QA samples.

The total batch size ($GPU\_number \times Batch\_size\_per\_device \times Gradient\_accumulation\_steps$) is 64. The learning rate increases linearly from 1e-6 to 2e-5 with 20 warm-up steps and remains constant in the following steps. The rank used in the LoRA (Hu et al., 2021) fine-tuning is set to 8. Following LongLoRA (Chen et al., 2024), the embedding and normalization layers are learnable during training. Moreover, we equip our method with FlashAttention-2 (Dao, 2023) for further acceleration and memory saving.

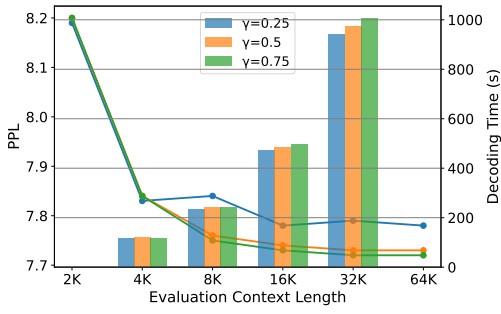 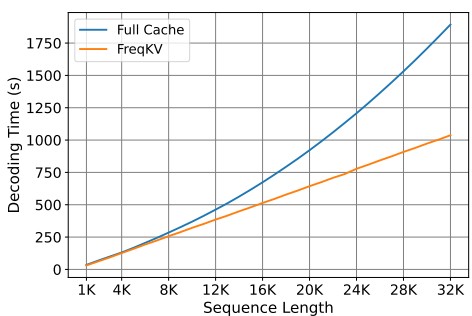

Figure 7: Perplexity (Line) and decoding latency (Bar) with different retaining ratios.

Figure 8: Decoding time with the full cache and FreqKV on the increasing sequence length.

## E    EVALUATION ON PROOF-PILE

In Table 9, we present the evaluation results on the Arxiv math dataset Proof-pile. On Proof-pile, FreqKV trails LongLoRA at longer context lengths. This gap arises because Proof-pile, unlike natural language corpora, contains dense symbolic and mathematical expressions where minor variations (e.g., a single symbol or index) drastically alter semantics. Such localized variations are more difficult to preserve under compression, making it particularly challenging for models using compressed cache to predict the next token.

LongLoRA retains the full KV cache during inference, avoiding any information loss and thus performing better on formula-heavy tasks. Nonetheless, FreqKV remains competitive and notably outperforms another compression-based method, LoCoCo. Additionally, the performance of FreqKV does not degrade further as the context length increases, highlighting its robustness.

## F    ABLATION STUDY

### F.1    RETAINING HIGH/LOW-FREQUENCY COMPONENTS

We further compare variants of FreqKV that retain only high-frequency vs. only low-frequency components using LLaMA-2-base-7B on the validation set of PG-19. Results (PPL) are provided in Table 10.

It shows that low-frequency components adapt better than the variant that retains high-frequency parts, which is consistent with our analysis in summarization tasks (Table 1). While the variant deteriorates when exceeding the original context window, low-frequency components, which carry more information, benefit the model, and the perplexity remains stable. This result aligns well with Frequency Principle (Xu et al., 2019; Luo et al., 2019) that low frequencies converge faster than high frequencies for Deep Neural Networks.

Table 10: Perplexity evaluation on PG-19 with high/low-frequency components retained.

| Components Retained | Evaluation Context Length | | | |
|---|---|---|---|---|
| | 2048 | 4096 | 8192 | 16384 |
| high-frequency | 8.29 | 7.92 | 8.42 | 8.70 |
| low-frequency | **8.20** | **7.84** | **7.74** | **7.73** |

### F.2    RETAINING RATIO

We train LLaMA-2-7B on RedPajama (Computer, 2023) with different retaining ratios. The training length is 8K. Models are evaluated on the validation set of PG-19 with the evaluation length ranging from 2K to 64K.

The performance of FreqKV with different retaining ratios is presented in Figure 7. The decoding time (seconds) is averaged over five runs. FreqKV performs better with more frequency components retained. However, larger retaining ratios lead to more frequent compression steps and denser attention computation, hence higher latency.

We also evaluate the performance of FreqKV when modifying its retaining ratio during evaluation on the validation set of PG-19. The PPL scores at a length of 8K are presented in Table 11. These results suggest that FreqKV is robust to mismatched training/evaluation retaining ratios, especially when using larger inference ratios than training (which retains more information).

Table 11: Perplexity evaluation on PG-19 with mismatched training/evaluation retaining ratios.

| Retaining Ratio during Training | Retaining Ratio during Evaluation | | |
|---|---|---|---|
| | 0.25 | 0.5 | 0.75 |
| 0.25 | 7.84 | 7.81 | 7.82 |
| 0.5 | 7.89 | 7.76 | 7.77 |
| 0.75 | 7.97 | 7.79 | 7.75 |

### F.3 SINK SIZE

The effect of sink token count has been explored in StreamingLLM (Xiao et al., 2024b). We also ablate the effect of the sink token number $S$ for FreqKV on LLaMA-2-7B. Results are given in Table 12. It shows that a threshold of four sink tokens appears to be enough, with subsequent additions contributing marginal effects.

Table 12: Perplexity evaluation on PG-19 with different numbers of sink tokens.

| Sink Size | Evaluation Context Length | | |
|---|---|---|---|
| | 2048 | 4096 | 8192 |
| 0 | 8.23 | 7.87 | 7.84 |
| 2 | 8.21 | 7.85 | 7.78 |
| 4 | 8.20 | 7.84 | 7.76 |
| 8 | 8.20 | 7.83 | 7.75 |

### F.4 EFFECT OF FREQUENCY-DOMAIN COMPRESSION

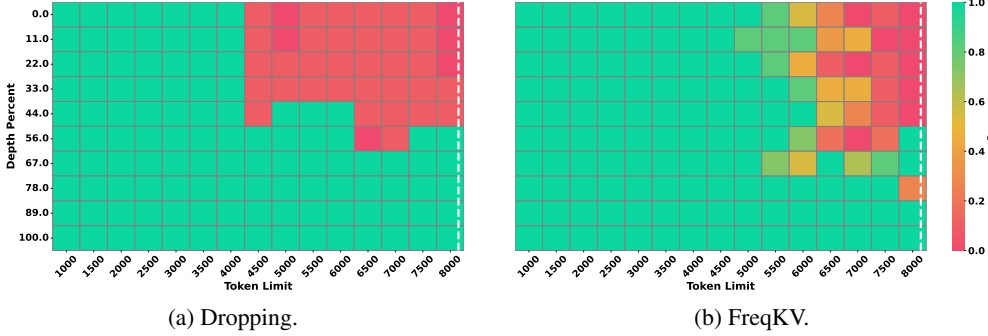

(a) Dropping.         (b) FreqKV.

Figure 9: Comparison of FreqKV and its variant Dropping on LLaMA-2-chat-7B.

To demonstrate the effectiveness of our compression operation, a variant method, Dropping, is also implemented and evaluated on Needle-in-a-Haystack for comparison, as in Figure 9. Instead of com-

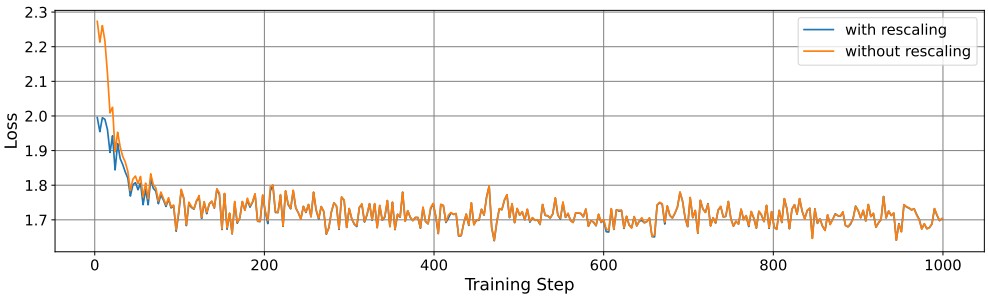

Figure 10: Curves of training loss for FreqKV with and without rescaling.

pressing in the frequency domain, Dropping retains the most recent tokens like StreamingLLM (Xiao et al., 2024b). Although trained under identical conditions, Dropping suffers from an irreversible loss of early-context information. FreqKV, by contrast, preserves essential signals and achieves superior performance.

### F.5 EFFECT OF RESCALING

In Equation 3, $\sqrt{\frac{L}{N}} = \sqrt{\gamma}$ works as a rescaling factor for the compressed signals to restore the original amplitude when conducting IDCT. In DCT, an $N$-length time-domain signal is normalized by $\sqrt{N}$; during IDCT, the retained $L$ frequency components are correspondingly normalized by $\sqrt{L}$. When iterative compression is applied across multiple steps, this mismatch in normalization scales can cause the reconstructed signals to be progressively amplified. In our pilot study, the training loss is significantly higher at the early stages when the compressed signals are not rescaled as presented in Figure 10. Therefore, we introduce the rescaling factor $\sqrt{\frac{L}{N}}$ to restore the original amplitude in the time domain.

## G COMPRESSION OVERHEAD

We conduct further studies on the compression overhead of FreqKV. FreqKV performs compressions on-the-fly during decoding. We compare the decoding time required by LLaMA-2-7B with the full cache and our FreqKV when the sequence length increases. As shown in Figure 8, the decoding time starts to diverge at the length of 4K. While the full cache utilization leads to a quadratic growth in decoding time, the decoding time of FreqKV increases approximately linearly, with a negligible time spent on compression, showcasing its efficiency.

As introduced in Section 3.4, chunk-wise attention is performed for the prefilling tokens. The size of the attention matrix in each chunk is $(N - L - S) \cdot N$ except for the last few tokens. Therefore, the computational cost of self-attention grows approximately linearly with the input length, like sliding window attention (Beltagy et al., 2020).

We use torchprofile[1] to count the number of Floating Point Operations (FLOPs) with input sequences of different lengths for LLaMA-2-7B. We calculate FLOPs for the model with the original attention, which leverages full KV states, and with FreqKV with the retaining ratio of 0.5. To quantify the compression overhead, we have measured FLOPs of Dropping, which retains the most recent tokens instead of compressing in the frequency domain. It shares the same sink size and retaining size as FreqKV. The difference in FLOPs between the two methods shows the overhead of compression.

The statistics are given in Table 13. The compression times of FreqKV with different context lengths are also reported in the table. It shows that the computation overhead of our compression process grows less than 0.5% even with a length of 32K, which could be negligible. FreqKV reduces more FLOPs as the input length grows from 4K to 32K compared to Full KV. This is because the compres-

---

[1]https://github.com/zhijian-liu/torchprofile

sion is performed every $N - L - S$ tokens with the complexity of $O(N\log N)$, which is negligible compared to the quadratic self-attention.

Table 13: FLOPs (TFLOPs) with input sequences of different lengths. Experiments are conducted on a single NVIDIA RTX6000 Ada-48GB GPU, where full KV cache of 12K tokens with float16 will cause an OOM (Out-of-Memory) issue. The difference between FreqKV and Dropping shows the computation overhead of compression.

| Models | 4K | 8K | 12K | 16K | 32K |
|---|---|---|---|---|---|
| Full Cache | 62.93 | 143.46 | OOM | OOM | OOM |
| Dropping | 62.93 | 125.86 | 188.79 | 251.72 | 503.43 |
| FreqKV | 62.93 | 125.90 | 188.85 | 251.81 | 503.63 |
| Compression Overhead | 0 (0%) | 0.039 (0.031%) | 0.064 (0.034%) | 0.090 (0.036%) | 0.193 (0.038%) |
| Compression Count | 0 | 3 | 5 | 7 | 15 |

