# OpenReview forum: "FreqKV: Key-Value Compression in Frequency Domain for Context Window Extension"
_ICLR.cc/2026/Conference — ICLR 2026 Poster_

### Official Review · Reviewer_phR3 · 2025-10-23

**Soundness:** 3
**Presentation:** 2
**Contribution:** 2
**Rating:** 4
**Confidence:** 3

**Summary:**

The authors propose a form of cache compression based on converting the tokens in the cache into the frequency domain for compressed storage. This results in lower storage requirements, lower attention complexity in decoding, and a kind of natural context extension since positional embeddings since the compression projects the cache into a fixed cardinality which will not expand to use up all of the positional embeddings.

**Strengths:**

- KV cache compression is an important topic due to the efficiency gains and power consumption concerns of modern transformers.

- In addition to compression the cache, the method also seems to cause an efficiency gain in the decoding attention operation.

**Weaknesses:**

- Llama 3 is used as a baseline model. This is important because I believe the only reason some baselines show poor performance is because they have exceeded the number of training positional embeddings. However, Llama 3 is already 1.5 years old at this point. There are already 3 releases of Llama3 which go up to 3.3 and have 131K native positional embeddings. Can FreqKV be applied to these models and show the same good performance past 131K?

- There is no comparison of latency with baselines such as SnapKV.

---

Overall, it would be more convincing if the authors could provide a case on what happens with extremely long contexts. Due to the compressive nature of the cache, it may only be able to hold information up to a certain point before the compression becomes noise. However, if the same trend witnessed at 8K-->16K can be witnessed on a 131K-->232K model, I would find this very compelling. This could be done with the exact same experimental setup and swapping in the Llama 3.x models.

**Questions:**

For a clearer understanding of how the DCT and IDCT matrices transform the inputs, can you add the dimensions for both the DCT and IDCT matrices?

---

> ### Author Response · Authors · 2025-11-19
> **Response to reviewer phR3**
>
> Dear reviewer, we sincerely appreciate your time and insightful comments. Here are our responses to your questions accordingly.
>
> ## 1. Performance at extremely long contexts
>
> While LLaMA-3.x models natively support 128K context lengths, many prior works on context extension continue to use LLaMA-2 as the backbone for fair comparison and accessibility. With 70B parameters, LLaMA-3.3 is beyond our compute budget. Therefore, we use LLaMA-3.2-1B, whose context window size is also 128K.
>
> Compared with LLaMA-2, LLaMA-3.2 has a vocabulary size of 128K--four times larger--so the token embedding layer consumes substantially more memory. To fit our hardware constraints, we fine-tuned LLaMA-3.2 with only 16K context length and an 8K cache during training. To test performance at longer contexts, we concatenate samples in PG-19 and evaluate up to 1M tokens. Perplexity scores at different lengths are reported in the following table.
>
> |Evaluation Length|128K|256K|384K|512K|640K|768K|896K|1M|
> |-|-|-|-|-|-|-|-|-|
> |log *PPL*|2.823|2.872|2.812|2.767|2.735|2.729|2.711|2.684|
>
> Despite being finetuned with the training length of 16K, FreqKV maintains stable performance when extrapolating beyond 128K. This behavior is consistent with our LLaMA-2 results in **Section 5.1 (Table 5)**, where FreqKV also preserves low perplexity as the evaluation length scales from 8K to 256K, far exceeding the training range. Overall, these results demonstrate that FreqKV generalizes well to extremely long contexts, even when applied to modern long-context backbones such as LLaMA-3.2.
>
> ## 2. Comparison of latency with baselines
>
> We compare latencies of SnapKV, PyramidKV, and FreqKV on LLaMA-2-7B with 4K tokens compressed to 2K. Since SnapKV and PyramidKV only compress KV cache during the prefill stage, we measure Time to First Token (TTFT) along with the compression latency. Results are averaged over 10 runs:
>
> |Methods|TTFT (s)|Compression latency (s)|
> |-|-|-|
> |SnapKV|0.8024|0.0382|
> |PyramidKV|0.8098|0.0456|
> |FreqKV|1.0489|0.2847|
>
> More latency is introduced in the compression process by FreqKV, primarily due to the Fast Fourier Transform (FFT) used for DCT/IDCT, which has computational complexity $O(N \log N)$.
>
> However, prefill accounts for only a small portion of total inference time in long-context generation. Unlike SnapKV and PyramidKV, FreqKV continues to compress KV cache throughout the decoding stage, allowing the KV cache to remain small at all time steps. As shown in **Table 5**,  the decoding time of FreqKV increases **approximately linearly**, with a negligible time spent on compression. The overhead of the compression accounts for **less than 0.3%** of the decoding latency and does not grow with the context length.
>
> ## 3. Matrix view of DCT and IDCT
>
> The mathematical definitions of DCT and IDCT are provided in **Appendix A**. Here, we present a matrix-form description for clarity. The transforms are applied **along the sequence dimension only**.
>
> Let $X \in R^{N \times d}$ be the original key/value sequence, where $N$ is the sequence length and $d$ is the feature dimension. The DCT transform can be written as:
> $$
> Z=DX,
> $$
>
> where $D\in R^{N\times N}$ is the orthonormal DCT matrix. To compress the KV cache, we keep only the lowest $L$ frequencies: $Z'=Z[:L] \in R^{L\times d}$. The IDCT matrix $\hat D\in R^{L \times L}$ will transpose the compressed components from the frequency domain to the time domain:
> $$
> X'=\hat DZ'
> $$
>
> We hope our responses have addressed your concerns and welcome any further discussion.

---

> > ### Comment · Reviewer_phR3 · 2025-11-27
> >
> > Thank you for your response. I think it is worth noting that there are some prior published works which also use the same style of iterative prefill with a cache compression mechanism such as [1,2]. Due to the similarity of the prefill strategy, I believe this should at least be mentioned as prior work.
> >
> > With regards to the added long context experiment, it is hard to place the significance of this result when there is nothing to compare it to, even a simple baseline like Streaming LLM [3], which would also stably stream on PG19 to these context lengths as shown by the original authors.
> >
> > Regarding the poor performance of baselines past their natively trained positional embeddings in Figure 3, the newly added results do not demonstrate that the performance of FreqKV can extend to this context length in the same manner as shown in Figure 3.
> >
> > Regarding the latency experiment, you say that the complexity of the FFT is $N \log N$ but isn't the transform only on a local block which essentially renders the complexity to a constant? Something like $O(N k \log k)$ with $N$ being the overall sequence length?
> >
> > ---
> >
> > [1] Infini Attention - https://arxiv.org/abs/2404.07143
> >
> > [2] Cascading Cache - https://arxiv.org/abs/2406.17808
> >
> > [3] Streaming LLM - https://arxiv.org/abs/2309.17453

---

> > > ### Author Response · Authors · 2025-11-28
> > > **Response to Reviewer phR3's follow-up**
> > >
> > > Dear Reviewer,
> > >
> > > Thanks for your response! We appreciate your feedback and have made every effort to address your questions thoroughly.
> > >
> > > ## 1. Suggestions on related work
> > >
> > > Thank you for pointing out these relevant works. Cascading Cache optimizes the eviction strategy of StreamingLLM in a cascaded manner, while Infini-Attention integrates compressive memory through a retrieval-based module. We will incorporate and discuss these methods in the revised manuscript.
> > >
> > > ## 2. Comparison with StreamingLLM
> > >
> > > We conduct a direct comparison between FreqKV and StreamingLLM on long-context language modeling. Results are shown below:
> > >
> > > |Evaluation Length|128K|256K|384K|512K|640K|768K|896K|1M|
> > > |-|-|-|-|-|-|-|-|-|
> > > |StreamingLLM (log *PPL*)|2.777|2.824|2.759|2.710|2.680|2.675|2.654|2.629|
> > > |StreamingLLM (decoding time/s)|4299.6|8582.6|12902.7|17309.0|21749.4|26235.4|30738.2|35220.2|
> > > |FreqKV (log *PPL*)|2.823|2.872|2.812|2.767|2.735|2.729|2.711|2.684|
> > > |FreqKV (decoding time/s)|4163.3|8344.5|12554.1|16791.7|21057.5|25342.6|29663.5|33992.0|
> > >
> > > These results show that both approaches maintain stable perplexity up to 1M tokens. However, it should be noted that the two methods differ fundamentally in their compression strategies. While StreamingLLM evicts one middle token per step, FreqKV compresses half of the KV tokens at once and performs compression only when the cache reaches capacity. Therefore, **FreqKV maintains a smaller effective cache size, resulting in slightly lower perplexity but faster decoding**.
> > >
> > > Although streaming-based methods have shown their effectiveness on language modeling, **they suffer from the information loss when dropping middle tokens**. Our experiments in **Appendix G.4** compare FreqKV with a streaming-based variant method, Dropping, on Needle-in-a-Haystack. Instead of compressing in the frequency domain, Dropping evicts middle tokens and retains the most recent tokens like StreamingLLM. Despite training under identical conditions, Dropping irreversibly loses early-context information, while FreqKV preserves information across all tokens via frequency-domain aggregation.
> > >
> > > ## 3. Context extension performance
> > >
> > > Figure 3 demonstrates that other KV compression methods fail beyond the original window boundary on Needle-in-a-Haystack. In contrast, FreqKV remains robust and demonstrates its effectiveness for context extension.
> > >
> > > Separately, our experiments on long context language modeling also show that FreqKV generalizes well in terms of length extrapolation. As shown in **Table 1** and **Table 5**, although the model is trained only up to 8K tokens, FreqKV maintains low perplexity as the evaluation length grows far beyond the training window, reaching up to 256K. Our additional results on LLaMA-3.2 further support the context extension capability of FreqKV.
> > >
> > > ## 4. Computation complexity of compression
> > >
> > > We are not sure whether $k$ in your notations stands for the context window size. To avoid confusion, we clarify the notation and the resulting complexity analysis.
> > >
> > > As stated in Line 256, $N$ stands for the maximum KV cache size, which is also the context window size. Employing FFT, the computation complexity of a single compression operation is $O(N \log N)$. The compression is triggered every $N-L$ tokens, with $L=\gamma N$ tokens retained.
> > >
> > > For a sequence of $N_S$ tokens, FreqKV will conduct compression for $\frac{N_S - N}{N-\gamma N}$ (ignoring constant-level rounding). Therefore, the total complexity is:
> > > $$
> > > O(\frac{N_S - N}{1-\gamma}\log N)
> > > $$
> > >
> > > For long sequences ($N_S \gg N$), this simplifies to:
> > > $$
> > > O(\frac{N_S}{1-\gamma}\log N)
> > > $$
> > >
> > > It shows that the per-compression cost is constant with respect to the overall sequence length, and the total complexity grows linearly with $N_S$, up to a small $\log N$ factor contributed by the FFT. Analysis for compression overhead is also provided in **Appendix H**.
> > >
> > > We hope that the above clarifications fully address your concerns.

---

### Official Review · Reviewer_35T2 · 2025-10-31

**Soundness:** 3
**Presentation:** 3
**Contribution:** 3
**Rating:** 6
**Confidence:** 4

**Summary:**

The paper proposed FreqKV, a novel KV cache compression method. FreqKV performs KV compression in the frequency domain using the DCT, retaining only the low-frequency components. After the fine-tuning, LLMs (Llama2 and Llama3) show comparable performance to the uncompressed one, and are superior to existing KV compression methods. The paper also reports generation speed improvements.

**Strengths:**

* The idea of applying frequency-domain compression to the KV cache is both intuitive and novel. The similarity to JPEG compression makes the concept easy to understand, and the empirical results demonstrate that this approach is competitive with or superior to existing baselines.
* The paper includes extensive comparisons with a wide range of prior methods on multiple datasets.

**Weaknesses:**

* The paper lacks a detailed analysis or intuitive explanation of why low-frequency components dominate in the KV cache. Moreover, the large magnitude of low-frequency components does not necessarily imply that they are semantically important, yet the paper seems to make this assumption. While the empirical evidence supports the method’s motivation and effectiveness, a more analytical approach would strengthen the argument.
* Although the authors claim speed improvements in both the prefill and decoding stages, the experiments seem to focus on decoding latency. The paper should provide either empirical results or additional clarification regarding the prefill stage speed-ups.
* In Table 3, competing methods fail completely (near zero point) at the 16K context length. This might not solely reflect a generalization failure but could be caused by an implementation difference: FreqKV applies pre-RoPE compression (on-the-fly RoPE just before inference), whereas others employ post-RoPE compression. It would be essential to clarify this difference.
* FreqKV requires a training phase for parameter learning; however, it remains unclear how it performs without such fine-tuning. Furthermore, comparisons in Table 3 should also include other training-based methods (e.g., LoCoCo).

**Questions:**

* The related work section would benefit from a broader discussion of recent streaming-based long-context KV management methods, such as InfLLM, InfiniPot, and Minference.
* The rescaling formulation (Eq. 5) is somewhat questionable. Instead of scaling by the number of retained coefficients, it might be more reasonable to adopt an (spectral) energy-preserving normalization.
* (minor) The abstract could be updated to refer to Llama 3 rather than Llama 2.

**Details Of Ethics Concerns:**

No concerns.

---

> ### Author Response · Authors · 2025-11-19
> **Response to reviewer 35T2 (part 1)**
>
> Dear reviewer, we sincerely appreciate your time and insightful comments. Here are our responses to your questions accordingly.
>
> ## 1. Further analysis of frequencies
>
> For time signals (e.g., texts) and space signals (e.g., images), their energy is most concentrated in low-frequency DCT components. This phenomenon occurs due to the mathematical properties of DCT and the inherent smoothness of real-world signals. The DCT expresses a signal as a sum of cosine functions oscillating at different frequencies. In the case of text embeddings, neighboring tokens exhibit strong temporal correlations, meaning adjacent values in the semantic space tend to be similar. Low-frequency cosine components are slowly varying and capture these smooth variations well, while high-frequency components mainly represent abrupt changes or noise.
>
> To empirically investigate what information different frequency bands encode, we have conducted a targeted perturbation experiment. We sample 100 CNN/DailyMail documents and introduce controlled perturbations by randomly replacing 1% of words with their synonyms. We then transform the KV states of LLaMA-2 into the frequency domain and compare the cosine similarity between perturbed and original inputs for both low-frequency (lowest 50%) and high-frequency (highest 50%) components. The results across different layers are summarized below:
>
> - key states
>
>     |layer|0|1|2|4|8|16|31|
>     |-|-|-|-|-|-|-|-|
>     |low frequencies|**0.8375**|**0.8799**|**0.8338**|**0.7932**|**0.7921**|**0.7938**|**0.8022**|
>     |high frequencies|0.3457|0.3526|0.3594|0.3563|0.3518|0.3490|0.3525|
>
> - value states
>
>     |layer|0|1|2|4|8|16|31|
>     |-|-|-|-|-|-|-|-|
>     |low frequencies|**0.6056**|**0.6056**|**0.6122**|**0.6423**|**0.6951**|**0.7268**|**0.7130**|
>     |high frequencies|0.3458|0.3390|0.3446|0.3497|0.3490|0.3484|0.3524|
>
> It shows that the **output states are robust to the perturbation with low-frequency components retained while high-frequency ones are more sensitive, supporting the interpretation that low frequencies encode global information, while high frequencies capture local details**.
>
> We further evaluate this distinction on the summarization track of the downstream benchmark LongBench, comparing the performance of LLaMA-2-chat-7b with low and high frequencies retained as below.
>
> |task|GovReport|QMSum|MultiNews|
> |-|-|-|-|
> |low-retained|**25.51**|**21.81**|**26.9**|
> |high-retained|14.21|16.54|15.55|
>
> Summarization requires capturing global semantic structure and long-range coherence. Accordingly, retaining low-frequency components yields substantially better performance.
>
> Together, these results strengthen our motivation: low-frequency components primarily encode global semantic contexts and long-range dependencies, whereas high-frequency components encode local details, which explains why preserving low-frequency information is both more robust and more beneficial for long-context understanding.
>
> ## 2. Computation cost during prefill stage
>
> As discussed in **Appendix H**, FreqKV performs chunk-wise attention in an iterative manner for the prefilling tokens. The computational cost for the prefill stage on LLaMA-2-7b is shown in the following table. The latency is averaged over 10 runs.
>
> |Evaluation Length|4K|8K|16K|
> |-|-|-|-|
> |Full Cache (latency/s)|0.5132|1.0086|OOM|
> |Full Cache (FLOPs/TFLOPs)|62.93|143.36|OOM|
> |FreqKV (latency/s)|0.5783|1.7599|3.6528|
> |FreqKV (FLOPs/TFLOPs)|62.93|125.90|188.85|
>
> Compressing contexts iteratively, FreqKV conducts self-attention chunk-by-chunk, which slows down the latency during the prefill stage. However, this iterative compression manner enables context window extension and avoids out-of-memory failure when full caching becomes infeasible (e.g., $\geq$ 16K tokens).
>
> It is also important to note that, in long-context generation workloads, **the decoding stage dominates total runtime**, rather than prefill. As shown in **Table 5**, the decoding latency of FreqKV increases approximately linearly, with a negligible time spent on compression. The overhead of the compression accounts for **less than 0.3%** of the decoding latency and does not grow with the context length. Thus, the additional prefill cost is amortized and remains practical for long-context applications.

---

> ### Author Response · Authors · 2025-11-19
> **Response to reviewer 35T2 (part 2)**
>
> ## 3. Benefits of pre-RoPE compression in FreqKV
>
> As discussed in **Section 4.3 (Line 365)**, most KV compression methods compress the long prompt after the first decoding step. Out-of-bound position embeddings are used when calculating attention, resulting in a performance breakdown. Different from these methods, **FreqKV first conducts self-attention within the context window, and then compresses the KV states for out-of-window tokens to get in**. Compressing key states before applying RoPE, FreqKV extends the context window without requiring position extrapolation or interpolation.
>
> Moreover, we ablate whether to compress key states before or after RoPE as in **Appendix D**, and observe that RoPE implicitly filters or distorts certain frequency components. As shown in **Table 8**, pre-RoPE compression consistently outperforms post-RoPE compression, reinforcing our design choice.
>
> ## 4. Performance of FreqKV without training
>
> We evaluate FreqKV on LLaMA-2-base-7b without additional training. Results (PPL) on the PG-19 validation set are reported below:
>
> |Evaluation Length|2K|4K|8K|16K|32K|
> |-|-|-|-|-|-|
> |w.o. training|8.25|7.87|8.31|8.43|8.49|
> |w. training|8.20|7.84|7.74|7.74|7.73|
>
> Without training, the performance drops when the evaluation length exceeds the original window size (4K). However, with minimal training of only 1000 steps, which is far less than pretraining, the perplexity remains stable even when the evaluation length is larger than the training length (8K).
>
> ## 5. Comparison with LoCoCo
>
> FreqKV is an efficient context extension method that iteratively compresses key-value states in the frequency domain. Our experiments were conducted following two aspects: Context Extension and KV compression. For context extension, we compare FreqKV with Full FT, LoCoCo, and LongLoRA on language modeling, which is commonly adopted by previous work.
>
> As for KV compression, performance on downstream tasks is measured following prior work. Although an average score on LongBench is reported by LoCoCo in the paper, they have not released evaluation scripts or details. We have implemented it by ourselves and got a much worse performance. Therefore, we did not include it in the paper.
>
> ## 6. Discussion of the rescaling factor
>
> We agree that an energy-preserving normalization is an appealing insight. However, in our experiments, restoring the full spectral energy by amplifying the retained low-frequency components caused training collapse. This is because the amplification introduces large deviations once transformed back into the time domain.
>
> As stated in **Line 199**, the rescaling factor arises from the normalization terms inherent in the DCT/IDCT transforms. In DCT, an $N$-length time-domain signal is normalized by $\sqrt N$; during IDCT, the retained $L$ frequency components are correspondingly normalized by $\sqrt L$. When iterative compression is applied across multiple steps, this mismatch in normalization scales can cause the reconstructed signals to be progressively amplified. In our pilot study, the training loss is significantly higher at the early stages when the compressed signals are not rescaled. Therefore, we introduce the rescaling factor $\sqrt{\frac{L}{N}}$ to restore the original amplitude in the time domain.
>
> ## 7. Suggestions on related work and abstract
>
> Thank you for your valuable suggestions. We will revise them in the paper.
>
> We hope our responses have addressed your concerns and welcome any further discussion.

---

> > ### Comment · Reviewer_35T2 · 2025-11-20
> >
> > Thank you for the response.
> >
> > Most of my concerns seem to be addressed. I appreciate the additional experiment on target-frequency perturbation, which strengthens the empirical support for the method.
> > It is also helpful to find out that the authors have already evaluated LoCoCo, spectral-energy-preserving variants, and related approaches. Please consider including these details in the revised manuscript so that readers can gain insights for future research.
> >
> > Based on these clarifications and additional results, I am raising my score from 6 to 8.

---

> > > ### Author Response · Authors · 2025-11-20
> > >
> > > We sincerely thank the reviewer for the thoughtful engagement with our rebuttal. We are pleased that the additional analyses and clarifications addressed your concerns. As suggested, we have incorporated the new discussions into the revised manuscript and highlighted the corresponding updates (in seagreen) for clarity. Thank you again for the constructive feedback and for the improved score.

---

### Official Review · Reviewer_DurU · 2025-11-01

**Soundness:** 2
**Presentation:** 2
**Contribution:** 3
**Rating:** 4
**Confidence:** 3

**Summary:**

authors propos FreqKV, a parameter-free, architecture-agnostic KV-cache compression method for LLMs. It applies a DCT along the sequence axis to KV tensors, keeping low-frequency components, then IDCTs back, with an iterative schedule that re-compresses older tokens as the cache grows.

**Strengths:**

1. Clear idea with strong intuition that low-frequency energy concentration in KV states

2. Comprehensive experiments on multiple benchmarks illustrated the effectiveness of the proposed method.

3. The ablation study provides more detailed information on frequency choice.

**Weaknesses:**

1. While the low-frequency concentration is a great motivation, the paper doesn't explore why this happens or what information is stored in which frequency bands. For instance, is the low-frequency "global context" and the high-frequency "local token-specific details"? Authors may provide a deeper analysis here to provide valuable insights

2. Attention heads, layers may carry different spectral content. A per-head adaptive $\gamma$ or power-based cutoff might outperform fixed ratios. Do the authors have ablation or adaptive strategies here?

3. Authors may need to compare the most recent baseline [1]

References:

[1] LaCache: Ladder-Shaped KV Caching for Efficient Long-Context Modeling of Large Language Models, ICML'25

**Questions:**

see. weaknesses

---

> ### Author Response · Authors · 2025-11-19
> **Response to reviewer DurU (part 1)**
>
> Dear reviewer, we sincerely appreciate your time and insightful comments. Here are our responses to your questions accordingly.
>
> ## 1. Further analysis of frequencies
>
> For time signals (e.g., texts) and space signals (e.g., images), their energy is most concentrated in low-frequency DCT components. This phenomenon occurs due to the mathematical properties of DCT and the inherent smoothness of real-world signals. The DCT expresses a signal as a sum of cosine functions oscillating at different frequencies. In the case of text embeddings, neighboring tokens exhibit strong temporal correlations, meaning adjacent values in the semantic space tend to be similar. Low-frequency cosine components are slowly varying and capture these smooth variations well, while high-frequency components mainly represent abrupt changes or noise.
>
> To empirically investigate what information different frequency bands encode, we have conducted a targeted perturbation experiment. We sample 100 CNN/DailyMail documents and introduce controlled perturbations by randomly replacing 1% of words with their synonyms. We then transform the KV states of LLaMA-2 into the frequency domain and compare the cosine similarity between perturbed and original inputs for both low-frequency (lowest 50%) and high-frequency (highest 50%) components. The results across different layers are summarized below:
>
> - key states
>
>     |layer|0|1|2|4|8|16|31|
>     |-|-|-|-|-|-|-|-|
>     |low frequencies|**0.8375**|**0.8799**|**0.8338**|**0.7932**|**0.7921**|**0.7938**|**0.8022**|
>     |high frequencies|0.3457|0.3526|0.3594|0.3563|0.3518|0.3490|0.3525|
>
> - value states
>
>     |layer|0|1|2|4|8|16|31|
>     |-|-|-|-|-|-|-|-|
>     |low frequencies|**0.6056**|**0.6056**|**0.6122**|**0.6423**|**0.6951**|**0.7268**|**0.7130**|
>     |high frequencies|0.3458|0.3390|0.3446|0.3497|0.3490|0.3484|0.3524|
>
> It shows that the **output states are robust to the perturbation with low-frequency components retained while high-frequency ones are more sensitive, supporting the interpretation that low frequencies encode global information, while high frequencies capture local details**.
>
> We further evaluate this distinction on the summarization track of the downstream benchmark LongBench, comparing the performance of LLaMA-2-chat-7b with low and high frequencies retained as below.
>
> |task|GovReport|QMSum|MultiNews|
> |-|-|-|-|
> |low-retained|**25.51**|**21.81**|**26.9**|
> |high-retained|14.21|16.54|15.55|
>
> Summarization requires capturing global semantic structure and long-range coherence. Accordingly, retaining low-frequency components yields substantially better performance.
>
> Together, these results strengthen our motivation: low-frequency components primarily encode global semantic contexts and long-range dependencies, whereas high-frequency components encode local details, which explains why preserving low-frequency information is both more robust and more beneficial for long-context understanding.

---

> ### Author Response · Authors · 2025-11-19
> **Response to reviewer DurU (part 2)**
>
> ## 2. Adaptive compression for heads or layers
>
> ### 2.1. Head-wise
>
> We agree that dynamic or head-wise retaining ratios are promising extensions. In **Appendix C**, we present head-wise spectral analyses and observe that all attention heads consistently exhibit strong energy concentration in low-frequency components. They exhibit similar distribution patterns in **Fig. 5**. Due to this stability and similarity across heads, we did not adopt differentiated per-head ratios in the current work.
>
> ### 2.2. Layer-wise
>
> As shown in **Fig. 1** and **Fig. 5**, the power spectrum distribution of KV states varies across layers. While the initial embeddings of natural languages in the first layer exhibit no strong low-frequency bias, deeper layers increasingly shift toward low-frequency dominance. This suggests that early layers may benefit from retaining more frequency components.
>
> We therefore experimented with assigning layer-specific retaining ratios based on the power distribution, while maintaining the overall retention budget as 50%. Results on LongBench are reported in the table below.
>
> |Model|NQA|Qasper|MF-en|HotpotQA|2WikiMQA|Musique|GovReport|QMSum|MultiNews|TREC|TriviaQA|SAMSum|LCC|RB-P|Avg.|
> |-|-|-|-|-|-|-|-|-|-|-|-|-|-|-|-|
> |LLaMA-2|18.7|19.2|36.8|25.4|32.8|9.4|27.3|20.8|25.8|61.5|77.8|40.7|52.4|43.8|35.17|
> |+FreqKV (adaptive)|20.23|16.43|**33.36**|34.5|33.98|10.4|**25.75**|**22.23**|**26.9**|**57.5**|**84.73**|40.96|58.37|54.63|37.14|
> |+FreqKV (fixed)|**20.41**|**21.05**|31.2|**34.54**|**34.78**|**14.47**|25.51|21.81|**26.9**|56.0|84.09|**41.53**|**58.99**|**58.65**|**37.85**|
>
> Although the adaptive strategy performs slightly better in several tasks like summarization, it fails to obtain consistent improvements across the benchmark. To keep the method clear and concise, we adopt the fixed-ratio strategy in the paper. However, it could be promising to explore adaptive strategies for different layers or other modules in future work.
>
> ## 3. Comparison with LaCache
>
> We compare the performance of FreqKV and LaCache on LongBench. Evaluation results on LLaMA-2 and LLaMA-3 with different retaining ratios are reported in the following table.
>
> |Model|NQA|Qasper|MF-en|HotpotQA|2WikiMQA|Musique|GovReport|QMSum|MultiNews|TREC|TriviaQA|SAMSum|LCC|RB-P|Avg.|
> |-|-|-|-|-|-|-|-|-|-|-|-|-|-|-|-|
> |LLaMA-2|18.7|19.2|36.8|25.4|32.8|9.4|27.3|20.8|25.8|61.5|77.8|40.7|52.4|43.8|35.17|
> |+LaCache (r=1%)|4.37|12.72|6.34|14.77|14.91|2.23|11.18|7.39|9.94|16.75|15.13|6.42|7.56|7.09|9.77|
> |+FreqKV (r=1%)|**19.44**|**20.2**|**31.68**|**35.89**|**32.0**|**12.52**|**15.96**|**17.39**|**15.91**|**60.5**|**83.98**|**33.07**|**62.47**|**56.57**|**35.54**|
> |+LaCache (r=50%)|16.27|16.55|**31.55**|32.62|26.22|7.72|22.84|20.34|25.16|**65.0**|83.28|38.46|47.97|43.41|34.10|
> |+FreqKV (r=50%)|**20.41**|**21.05**|31.2|**34.54**|**34.78**|**14.47**|**25.51**|**21.81**|**26.9**|56.0|**84.09**|**41.53**|**58.99**|**58.65**|**37.85**|
> |LLaMA-3|22.52|31.83|41.04|44.14|36.68|24.05|28.87|23.25|26.46|75.5|90.23|42.09|56.7|51.84|42.51|
> |+LaCache (r=1%)|14.27|5.87|20.73|40.86|33.07|18.58|15.01|7.76|10.89|40.0|77.92|16.23|40.7|40.51|27.31|
> |+FreqKV (r=1%)|**22.86**|**39.7**|**36.44**|**44.29**|**39.59**|**22.57**|**29.1**|**22.92**|**28.35**|**65.0**|**89.0**|**41.9**|**62.17**|**55.42**|**42.81**|
> |+LaCache (r=50%)|22.95|28.91|40.99|46.34|36.26|**23.02**|16.92|16.02|15.59|**74.0**|**91.11**|32.97|55.09|49.31|39.25|
> |+FreqKV (r=50%)|**23.14**|**41.6**|**43.62**|**46.48**|**40.54**|20.2|**29.99**|**23.67**|**28.58**|69.0|88.95|**41.47**|**60.33**|**51.11**|**43.48**|
>
> **FreqKV consistently outperforms LaCache across most tasks on the two models**. Even with an extremely low retaining ratio (r = 1%), where LaCache degrades sharply, FreqKV remains stable and effective, highlighting its robustness under aggressive compression.
>
> We hope our responses have addressed your concerns and welcome any further discussion.

---

> > ### Comment · Reviewer_DurU · 2025-11-20
> > **Thank you**
> >
> > Thanks for addressing my concerns. I have raised my score.

---

> > > ### Author Response · Authors · 2025-11-20
> > >
> > > We sincerely thank the reviewer for the thoughtful engagement with our rebuttal. We are pleased that the additional analyses and clarifications addressed your concerns. We have incorporated the new discussions into the revised manuscript and highlighted the corresponding updates (in seagreen) for clarity. Thank you again for the constructive feedback and for the improved score.

---

### Comment · Area_Chair_jj77 · 2025-11-20
**Action Needed: Review Rebuttal and Update Evaluation**

Dear Reviewers,

Thank you, as always, for your valuable contributions and efforts. The authors have now submitted their rebuttal. Please take a moment to review it and provide any necessary follow-up actions, such as additional questions, clarification requests, or updates to your review.

Since the initial ratings ranged from 8 to 4, I kindly ask you to pay close attention to the perspectives of the other reviewers when preparing your final response.

Thank you again for your support.

---

### Author Response · Authors · 2025-12-01
**Summary of Rebuttal**

Dear Area Chair,

Thank you very much for taking over the assessment of our submission under the special circumstances. We sincerely appreciate your additional effort. To facilitate your quick understanding of the paper and our rebuttal, we provide a concise summary below.

## Key strengths highlighted by reviewers

- **Simple, intuitive, and novel idea**: Reviewers praised the use of DCT-based frequency-domain compression for KV cache as both intuitive and effectively motivated.

- **Architecture-agnostic and parameter-free**: FreqKV requires no architectural modifications and introduces no new learnable parameters, making it lightweight, clean, and easy to integrate into existing LLMs.

- **Efficiency**: Reviewers acknowledged the minimal computational overhead of the method, especially the near-linear decoding-time scaling and very small compression cost.

- **Comprehensive and convincing experiments**: The paper includes extensive experiments across LLaMA-2 and LLaMA-3 variants, covering language modeling, LongBench, RULER, and NIAH, along with thorough ablations and analyses.

We have carefully prepared our rebuttal to address all concerns through additional analyses, expanded experiments, and clearer explanations.

## Main questions and our responses

- **Further analysis of frequencies**: We conducted controlled perturbation and summarization experiments to show that low frequencies are far more robust and capture global information while high frequencies capture local details. (*Reviewer DurU and 35T2*)

- **Adaptive compression for heads or layers**: We provided experimental analysis of adaptive compression, and showed that the adaptive strategy does not obtain consistent improvements across the benchmark, which supports our choice of a simple global retention ratio. (*Reviewer DurU*)

- **Additional baseline**: We added direct comparisons with LaCache, demonstrating consistent advantages across models and retaining ratios. (*Reviewer DurU*)

- **Latency and complexity analysis**: We measured computational cost during prefill stage, which accounts for only a small portion of total inference time in long-context generation. We clarified overall computational complexity and confirmed that decoding latency scales approximately linearly with negligible compression overhead. (*Reviewer 35T2 and phR3*)

- **Benefits of pre-RoPE compression in FreqKV**: We explained that the iterative pre-RoPE compression in FreqKV avoids out-of-bound positional extrapolation. (*Reviewer 35T2*)

- **Performance without training**: We evaluated FreqKV without training and showed that with minimal training, the perplexity remains stable even when the evaluation length exceeds the training window. (*Reviewer 35T2*)

- **Applicability to LLaMA-3.2 and extremely long contexts**: We applied FreqKV on LLaMA-3.2 (fine-tuned at 16K) and evaluated up to 1M tokens, showing stable perplexity and competitive decoding latency. (*Reviewer phR3*)

- **Additional clarification**: We provided further discussions on the comparison with LoCoCo, the effect of the rescaling factor, the matrix view of DCT/IDCT, and context extension performance. (*Reviewer 35T2 and phR3*)

After our added analyses, **Reviewer DurU has raised the score from 4 to 6 (20/11/2025), and Reviewer 35T2 has raised the score from 6 to 8 (20/11/2025). Reviewer phR3 also provided further feedback on 27/11/2025, to which we responded with additional discussion on 28/11/2025.**

We are grateful to all Reviewers for engaging in the discussion and for acknowledging that our rebuttal addressed their questions. We sincerely appreciate Chair's time and effort in assessing our submission, and we are happy to provide further clarification if anything is unclear.

---

### Meta-Review · Area_Chair_sFNh · 2026-01-12

**Summary:**

The reviewers largely agree that FreqKV presents a simple, intuitive, and technically sound approach to KV cache compression by operating in the frequency domain, and that the method is unusually clean in being parameter-free and architecture-agnostic. The core concerns that initially limited enthusiasm were not about correctness, but about depth of understanding and positioning: several reviewers questioned whether the low-frequency motivation was merely heuristic or truly informative, whether adaptive strategies across heads or layers were necessary, whether the claimed efficiency held in both prefill and decoding stages, and whether comparisons against recent baselines (e.g., LaCache, SnapKV, StreamingLLM, Infini-Attention–style methods) were sufficiently comprehensive. There were also concerns about whether the apparent context extension gains were partly an artifact of pre-RoPE compression choices, and whether the method would continue to behave sensibly at extremely long contexts or on more modern backbones.

**Reviewer Concerns:**

The rebuttal was exceptionally strong and directly addressed essentially all substantive technical concerns with concrete evidence. The authors provided targeted perturbation experiments and downstream analyses that convincingly support the interpretation that low-frequency components encode global, robust semantic information, while high-frequency components capture more local and fragile details. They explored head-wise and layer-wise adaptive strategies and showed that, while sometimes helpful, they do not yield consistent gains, justifying the design choice of a fixed global retention ratio. The rebuttal added direct comparisons with LaCache, SnapKV, PyramidKV, StreamingLLM, and clarified the role and benefit of pre-RoPE compression with ablations. Importantly, the authors extended experiments to extremely long contexts (up to 1M tokens) and to LLaMA-3.2, showing stable perplexity and competitive decoding latency, which directly resolves prior doubts about scalability and modern relevance. What remains outstanding is relatively minor: prefill latency is indeed higher than eviction-based methods, and while the authors convincingly argue this cost is amortized in long-context generation, the method is clearly optimized for decode-heavy workloads rather than all inference regimes. These are scope limitations rather than unresolved flaws.

**Reviewer Scores:**

Reviewer DurU initially scored the paper below the acceptance threshold, raising thoughtful questions about the semantic meaning of frequency components, adaptive compression, and missing baselines. This reviewer actively participated in the discussion, explicitly acknowledged that the added analyses addressed their concerns, and raised their score from 4 to 6.

Reviewer 35T2 initially gave a marginally positive score and raised concerns about the interpretation of low-frequency dominance, prefill latency, and pre- vs. post-RoPE compression. This reviewer engaged deeply in the discussion, explicitly stated that the rebuttal addressed most concerns, and raised their score from 6 to 8, providing a clear signal of increased confidence.

Reviewer phR3 initially scored the paper below threshold, with concerns about applicability to newer LLaMA-3 variants, lack of latency comparison to certain baselines, and the significance of extreme-length experiments. This reviewer participated in follow-up discussion, and the authors responded with additional long-context experiments, direct StreamingLLM comparisons, and clarified complexity analysis. While phR3 did not explicitly update their score, the AC judges that the rebuttal materially addressed the core of their concerns and would plausibly move the reviewer closer to the acceptance boundary.

---

### Decision · Program_Chairs · 2026-01-26

Accept (Poster)